# Data Dependent Regret Bounds for Online Portfolio Selection with Predicted Returns

**Sudeep Raja Putta**    SP3794@COLUMBIA.EDU  and  **Shipra Agrawal**    SA3305@COLUMBIA.EDU
*Columbia IEOR*

**Editors:** Gautam Kamath and Po-Ling Loh

## Abstract

We study data-dependent regret bounds for the Online Portfolio Selection (OPS) problem. As opposed to worst-case bounds that hold uniformly over all sequence of returns, data-dependent bounds adapt to the specific sequence of returns seen by the investor. Consider a market of $n$ assets and $T$ time periods, consisting of the returns $r_1, \ldots, r_T \in \mathbb{R}_+^n$. The regret of our proposed algorithm, Log-Barrier Adaptive-Curvature Online Newton Step (LB-AdaCurv ONS) is bounded by $O(\min(nR \log T, \sqrt{nT \log T}))$, where $R = \max_{t,i,j} \frac{r_t(i)}{r_t(j)}$ is a data dependent quantity that is not known to the algorithm. Thus, LB-AdaCurv ONS has a worst case regret of $O(\sqrt{nT \log T})$ while simultaneously having a data-dependent regret of $O(nR \log T)$.

Next, we consider the more practical setting of OPS with predicted returns, where the investor has access to predictions that can be incorporated into the portfolio selection process. We propose the Optimistic Expected Utility LB-FTRL (OUE-LB-FTRL) algorithm that incorporates the predictions using a utility function. If the predictions are accurate, OUE-LB-FTRL's regret is $O(n \log T)$ and even if the predictions are arbitrary, regret is always bounded by $O(\sqrt{nT \log T})$. We provide a meta-algorithm called Best-of-Both Worlds for OPS (BoB-OPS), that combines the portfolios of an expected utility investor and a regret minimizing investor. By properly instantiating BoB-OPS, we show that the regret with respect to the expected utility investor is $O(\log T)$ and the static regret is $O(n \log T)$.

Finally, we also show new First-Order, Second-Order and Gradual-Variation regret bounds for OPS. In our analysis, we developed new regret inequalities for optimistic FTRL with convex hint functions. Our framework extends prior work on optimistic FTRL that only used linear hints, and so could be of independent interest.

**Keywords:** Online Portfolio Selection, Data-dependent regret bounds, Online Newton step, Follow the regularized leader, Optimistic online convex optimization.

## 1. Introduction

The online portfolio selection problem (OPS), as formulated by Cover (1991), is a repeated game of sequential investment between an investor (the player) and a market (the environment) consisting of $n$ assets (stocks). The investor starts with 1 unit of wealth. At the start of each investment period, indexed by $t = 1, 2, \ldots, T$, the investor distributes her wealth among the $n$ assets according to a *portfolio* vector $w_t$ belonging to the unit simplex $\Delta_n$, which is the set $\{w \in \mathbb{R}^n : \sum_{i=1}^n w(i) = 1, w(i) \geq 0, i \in [n]\}$. At the end of the investment period, the investor observes non-negative *returns* $r_t \in \mathbb{R}_+^n$ from the market and her wealth changes by a multiplicative factor of $r_t^\top w_t$. The wealth after $T$ periods will be $\prod_{t=1}^T (r_t^\top w_t)$.

---

**Online Portfolio Selection - Interaction Protocol:**

**for** $t = 1$ *to* $T$ **do**

$\quad$ Investor picks the portfolio $w_t \in \Delta_n$ based on past returns $\{r_1, \ldots, r_{t-1}\}$

$\quad$ Market revels $r_t \in \mathbb{R}_+^n$

$\quad$ Investor's wealth grows by a multiplicative factor of $r_t^\top w_t$

**end**

---

The wealth of an investor that selects the same portfolio $w$ in each period is $\prod_{t=1}^T (r_t^\top w)$. We compare the difference in the log-wealth between the two investors:

$$\log \left( \prod_{t=1}^T (r_t^\top w) \right) - \log \left( \prod_{t=1}^T (r_t^\top w_t) \right) = \sum_{t=1}^T (-\log(r_t^\top w_t)) - (-\log(r_t^\top w))$$

If we define $f_t(w) = -\log(r_t^\top w)$ as our convex cost function, then the above expression exactly corresponds to the notion of static regret $\sum_{t=1}^T f_t(w_t) - f_t(w)$, which is the main object of study in Online Convex Optimization (OCO) ([Hazan, 2016](#)). Prior works on OPS have mainly focused on bounding the *worst-case* regret of algorithms for the OPS problem. Instead, we study various *data-dependent* regret bounds that bound the regret of the algorithm in terms of the market returns.

In the classical version of the OPS problem, it is assumed that the investor does not have any prior belief about the future returns. At time $t$, the investor only uses the returns $r_1, \ldots, r_{t-1}$ and possibly prior portfolios $w_1, \ldots, w_{t-1}$ to select the portfolio vector $w_t$. However, in practice, investors do have prior beliefs about future returns. This prior belief is typically expressed as a distribution $D_t$ over the future returns of assets, i.e., the investor believes that $r_t$ would be a random variable drawn from $D_t$. Thus the distribution $D_t$ is the investor's prediction for the current return. The actual market return $r_t$ however, could be arbitrary.

---

**Online Portfolio Selection with Predictions - Interaction Protocol:**

**for** $t = 1$ *to* $T$ **do**

$\quad$ Investor receives prediction $D_t$ distribution.

$\quad$ Investor picks the portfolio $w_t \in \Delta_n$ based on past returns $\{r_1, \ldots, r_{t-1}\}$ and $D_t$

$\quad$ Market revels $r_t \in \mathbb{R}_+^n$

$\quad$ Investor's wealth grows by a multiplicative factor of $r_t^\top w_t$

**end**

---

We study algorithms that incorporate the investor's predictions to further reduce regret. As such, the specific process by which an investor creates these predictions in each round is not the subject of our study. Instead, we study how an investor could incorporate such predictions into the online portfolio selection framework.

### 1.1. Notation

Let $\Delta_n$ be the probability simplex $\{w \in \mathbb{R}^n : \sum_{i=1}^n w(i) = 1, w(i) \geq 0, i \in [n]\}$. The all ones vector are denoted by $\mathbf{1}$. The set of real numbers is $\mathbb{R}$, non-negative numbers is $\mathbb{R}_+$ and positive

numbers is $\mathbb{R}_{++}$. The Hadamard product of two vectors $u, v$ is represented as $u \circ v$. The Bregman Divergence of function $F$ is $\mathrm{B}_F(x\|y) = F(x) - F(y) - \nabla F(y)^\top (x - y)$.

## 2. Prior Works and Our Contributions

### 2.1. Worst-Case Regret Bounds

In his seminal work, Cover (1991) proposed the OPS problem and the Universal Portfolio (UP) algorithm. The minimax regret for OPS is $\Theta(n \log T)$ and UP obtains this rate. Helmbold et al. (1998) identified that the OPS can be posed as an OCO problem and propose using the Exponentiated Gradient (EG) (Kivinen and Warmuth, 1997). Since the EG algorithm requires the gradient of $f_t$ to be bounded, it results in a regret bound that is not uniform over the sequence of returns. Their result states that for any sequence of returns $r_1, \ldots, r_t \in [c, C]^n$, the regret of EG is $O(C/c\sqrt{T \log n})$. Here $c, C \in \mathbb{R}_{++}^n$ are assumed to be known apriori by the investor as they are used to tune parameters within the EG algorithm. Helmbold et al. (1998) also propose a technique to "universalize" EG and obtain a regret of $O(n^{1/2}T^{3/4})$ for any sequence of returns $r_1, \ldots, r_T \in \mathbb{R}_+^n$. This bound was later improved to $O(n^{1/3}T^{2/3})$ in Tsai et al. (2023a).

Hazan et al. (2007) propose the Online Newton Step(ONS) method. With apriori knowledge of $c, C$, ONS has $O\left(C/c \, n \log T\right)$ regret for all sequences of returns in $[c, C]^n$. Using the universalization technique (Helmbold et al., 1998), it is possible to modify ONS to obtain a regret of $O(n\sqrt{T \log T})$ for all return sequences in $\mathbb{R}_+^n$. There have been several recent works that study worst case regret-bounds for OPS. We summarize these algorithms in Table 5.

### 2.2. Data-dependent Regret Bounds

In this paper, we aim to study algorithms with data-dependent regret bounds for OPS. Possibly the most simple algorithm for OPS is Follow-The-Leader (FTL). The iterates of FTL are $w_t \in \arg\min_{w \in \Delta_n} \sum_{s=1}^{t-1} f_t(w)$. Agarwal and Hazan (2006) introduce the Smooth Prediction (SP) algorithm that adds the log-barrier regularizer to FTL. A similar algorithm called Exp-Concave FTL (Hazan and Kale, 2015), uses the $\ell_2$-regularizer instead of the log-barrier.

There are no parameters to tune in these algorithms. All three of these can be shown to possess the following data-dependent regret bound. For any sequence of returns $r_1, \ldots, r_T \in \mathbb{R}_+^n$, the regret of FTL, SP, and Exp-Concave FTL is $O(R^2 n \log T)$, where $R = \max_{t, i, j} \frac{r_t(i)}{r_t(j)}$ is the data-dependent quantity. While in the worst-case, $R$ could be unbounded, it nevertheless yields reasonable regret bounds for benign sequences of returns $r_1, \ldots, r_T$.

Using an adaptive variant of EG called the AdaHedge algorithm (van Erven et al., 2011; de Rooij et al., 2014), one can obtain a regret of $O(R\sqrt{T \log n})$ for any sequence of returns. Using so-called *universal online convex optimization* (UOCO) algorithms such as Metagrad (van Erven et al., 2021) and Maler (Wang et al., 2019), one can obtain a regret bound of $O(Rn \log T)$. However, UOCO algorithms function by running $O(\log T)$ instances of ONS instantiated with different parameters and running a meta experts algorithm to control them.

We show that a simple variant of the ONS algorithm that employs *adaptive-curvature surrogate functions* has $O(Rn \log T)$ regret. We term this algorithm AdaCurv ONS. Thus, we can avoid the use of complicated UOCO algorithms like Metagrad (van Erven et al., 2021) and Maler (Wang et al., 2019). However, all the above algorithms have unbounded regret in the worst case, due to the dependence on $R$. For some sequences of returns, $R$ could be arbitrarily large. To avoid this

Table 1: Data-dependent Regret Bounds for Online Portfolio Selection

| Algorithm | Data-dependent Regret | Worst-case Regret | Run-time |
|---|---|---|---|
| FTL, SP (Agarwal and Hazan, 2006), Exp-Concave FTL (Hazan and Kale, 2015) | $R^2 n \log(T)$ | $\infty$ | $n^{2.5} T$ |
| AdaHedge (van Erven et al., 2011; de Rooij et al., 2014) | $R\sqrt{T \log n}$ | $\infty$ | $n$ |
| Metagrad (van Erven et al., 2021) Maler (Wang et al., 2019) | $Rn \log(T)$ | $\infty$ | $n^3 \log T$ |
| AdaCurv FTAL /ONS (Theorem 2) | $Rn \log(T)$ | $\infty$ | $n^3$ |
| LB-AdaCurv FTAL/ONS (Theorem 3) | $Rn \log(T)$ | $\sqrt{nT \log T}$ | $n^3$ |

issue, we propose adding the log-barrier regularizer along with an adaptivley tuned learning rate to AdaCurv ONS, obtaining a regret bound of $O(\min(Rn \log T, \sqrt{nT \log T}))$. We term this algorithm LB-AdaCurv ONS. These prior works, along with our contributions are summarized in Table 1.

## 2.3. OPS with Predicted Returns

In the OPS with predicted returns framework, the investor could leverage predictions to pick a portfolio in several possible ways. In his seminal work, which birthed the field of *Modern Portfolio Theory*, Markowitz (1952) proposed using an optimization framework that balances the expected return of a portfolio with its variance. Given a predicted returns distribution $D_t$, Markowitz's approach of mean-variance optimization can be stated as $w_t \in \arg\max_{w \in \Delta_n} \mathbb{E}_{r \sim D_t}[r^\top w] - \frac{\lambda}{2} \mathbb{V}ar_{r \sim D_t}[r^\top w]$. Here $\lambda$ is the "risk-aversion" parameter. *Capital Growth Theory*, developed by Kelly (1956); Thorp (1975), proposes to optimize the "Kelly criteria", $w_t \in \arg\max_{w \in \Delta_n} \mathbb{E}_{r \sim D_t}[\log(r^\top w)]$. Both of Markowitz's and Kelly's approaches are special cases of *Expected Utility Theory*. Given an investor with a specified concave utility function $U$ and return prediction $D_t$, the investor picks $w_t$ as $w_t \in \arg\max_{w \in \Delta_n} \mathbb{E}_{r \sim D_t}[U(r^\top w)]$.

Unlike OPS techniques, expected utility theory does not provide worst-case or data-dependent performance guarantee for the investor. On the other hand, an expected utility investor will outperform a regret minimizing investor if the predictions are indeed accurate. We hope to combine the two approaches of expected utility maximization and regret minimization, while maintaining their advantages. We initiate the study of online portfolio selection algorithms with predicted returns. At time $t$, these algorithms not only use the returns $r_1, \ldots, r_{t-1}$ and prior portfolios $w_1, \ldots, w_{t-1}$, but also use any available return predictions $D_t$ to select the portfolio vector $w_t$. We introduce algorithms that aim to optimize portfolio selection by balancing the inherent trade-off between exploiting these predictions to improve performance and maintaining robustness against prediction inaccuracies.

Online decision making algorithms that incorporate predictions have been studied under the name of *Online Learning with Predictions* (OLP) (Rakhlin and Sridharan, 2013a) in the online

Table 2: First-Order Regret Bounds for Online Portfolio Selection

| Algorithm | First-Order Regret | Worst-case Regret | Run-time |
|---|---|---|---|
| UOCO Yan et al. (2023) | $Rn \log L_T^\star$ | $\infty$ | $n^3 \log T$ |
| Tsai et al. (2023b) | $\sqrt{nL_T^\star} \log T$ | $\sqrt{nT} \log T$ | $n$ |
| AdaCurv ONS (Theorem 6) | $Rn \log(L_T^\star)$ | $\infty$ | $n^3$ |
| LB-AdaCurv ONS ( Theorem 7) | $\min(Rn \log(L_T^\star), \sqrt{nL_T^\star \log T})$ | $\sqrt{nT \log T}$ | $n^3$ |

optimization literature, and *Algorithms with Predictions* (Mitzenmacher and Vassilvitskii, 2022) in the online algorithms literature. Algorithms that incorporate predictions seek to achieve some form of *consistency* and *robustness* guarantees. Consistency implies that the algorithm should be able to improve its performance by taking advantage of the predictions in case they are accurate. Robustness implies that the algorithm should retain its worst-case performance guarantee in case the predictions have large errors or are misspecified. In the context of OPS, we present two algorithms that achieve different consistency and robustness guarantees.

We present the Optimistic Expected Utility LB-FTRL (OUE-LB-FTRL) algorithm that achieves a worst case static-regret of $O(n \log T)$ when the predictions are exact and $O(\sqrt{nT \log T})$ when the predictions are completely arbitrary. In other words, we say that this algorithm is $O(n \log T)$-consistent and $O(\sqrt{nT \log T})$-robust with respect to static regret. In the presence of predictions, it is important to also consider the regret with respect to the expected utility investor instead of static regret. We propose the Best-of-Both Worlds for OPS (BoB-OPS) algorithm that combines the portfolios of an expected utility investor and a regret minimizing investor to achieve simultaneously a $O(\log T)$ regret against the expected utility investor and $O(n \log T)$ static regret.

## 2.4. More Data-Dependent Regret Bounds

Finally, we study three more kinds of data-dependent regret bounds that have been previously considered in the literature.

### 2.4.1. FIRST-ORDER REGRET BOUND

Let $L_T^\star = \min_{w \in \Delta_n} \left[ \sum_{t=1}^T f_t(w) \right] - \sum_{t=1}^T \left[ \min_{w \in \Delta_n} f_t(w) \right]$. In the worst case, $L_T^\star = O(T)$. However, if the returns are not adversarial generated, then $L_T^\star$ could be quite small. Bounds which depend on $L_T^\star$ instead of $T$ are termed First-Order bounds. These are also called $L^\star$ bounds or Small-Loss bounds.

Orabona et al. (2012) show that ONS has the regret bound $O((C/c)n \log L_T^\star)$, when the returns are known to be in $[c, C]^n$. The recent UOCO algorithm from Yan et al. (2023) has a regret of $O(Rn \log L_T^\star)$ for any sequence of returns. However, these do not guarantee a bounded worst-case regret. Tsai et al. (2023b) were able to obtain a bound of $O(\sqrt{dL_T^\star} \log T)$ for any sequence of returns, with a run time of $O(n)$. We show that the AdaCurv ONS algorithm achieves $O(Rn \log L_T^\star)$ regret. Moreover, by adding extra regularization via the log-barrier, the LB-AdaCurv ONS algorithm achieves a regret of $O(\min(Rn \log(L_T^\star), \sqrt{nL_T^\star \log T}))$. These results are summarized in Table 2.

Table 3: Second Order Regret Bounds for Online Portfolio Selection

| Algorithm | Second Order Regret | Worst-case Regret | Run-time |
|---|---|---|---|
| Exp-Concave FTL (Hazan and Kale, 2015) | $R^2 n \log(Q_T)$ | $\infty$ | $n^{2.5} T$ |
| Tsai et al. (2023b) | $\sqrt{n \tilde{Q}_T \log T}$ | $\sqrt{nT \log T}$ | $n \log \log T$ |
| AdaCurv ONS ( Theorem 8) | $R^2 n \log(Q_T)$ | $\infty$ | $n^3$ |

### 2.4.2. SECOND-ORDER REGRET BOUND

Second-Order bounds are also known as Quadratic variation bounds. Hazan and Kale (2015) show that the Exp-Concave FTL algorithm has a regret of $O(R^2 n \log Q_T)$, where $Q_T = \min_x \sum_{t=1}^{T} \|r_t - x\|_2^2$. They also proposed an algorithm that uses a quadratic surrogate, called Faster Quadratic-variation Universal Algorithm (FQUA). This has a regret bound of $O((C/c)^3 n \log Q_T)$ when the returns are known to be in $[c, C]^n$. Recently, Tsai et al. (2023b), showed a second order bound $O(\sqrt{n \tilde{Q}_T \log T})$, where $\tilde{Q}_T = \min_x \sum_{t=1}^{T} \left\| \frac{r_t \circ w_t}{r_t^\top w_t} - x \right\|_2^2$ with a run time of $O(n \log \log T)$ for any sequence of returns. Since $\tilde{Q}_T = O(T)$, their algorithm has a worst-case regret bound of $O(\sqrt{nT \log T})$. However, $\tilde{Q}_T$ has no meaningful interpretation in terms of quadratic variation of returns $r_t$. We show that AdaCurv ONS has $O(Rn \log Q_T)$ regret. These results are summarized in Table 3.

### 2.4.3. GRADUAL-VARIATION BOUND

*Gradual-Variation Bounds* are data dependent regret bounds where the regret is bounded as a measure of variation between consecutive returns. These are also called Path-Length bounds. For the OPS problem, these bounds were first studied in Chiang et al. (2012), who showed that an optimistic variant of the ONS algorithm has a regret of $O(nC^2 \log V_T)$ when the returns are known to be in $[c, C]^n$. Here $V_T = \sum_{t=1}^{T} \|r_t - r_{t-1}\|_2^2$. The UOCO algorithm of Yan et al. (2023) obtains a gradual variation bound of the form $O(nR \log V_T)$. Due to the dependence on $R$ the worst-case regret of this approach is not bounded.

Tsai et al. (2023b) obtain a regret bound of $O(\sqrt{n \tilde{V}_T \log T})$, where $\tilde{V}_T = \sum_{t=2}^{T} \left\| \frac{r_t \circ w_{t-1}}{r_t^\top w_{t-1}} - \frac{r_{t-1} \circ w_{t-1}}{r_t^\top w_{t-1}} \right\|_2^2$. It implies a worst case $O(\sqrt{nT} \log T)$ regret bound. Their algorithm is an instance of log-barrier FTRL and uses *multiplicative-gradient optimism*, which is an implicit technique for simultaneously guessing the gradient $\nabla f_t(w_t)$ and picking the portfolio $w_t$. Their algorithm does not allow for specifying a utility function or predicted distribution like our OUE-LB-FTRL algorithm.

We obtain a gradual-variation bound of $O(\sqrt{n \tilde{V}'_T \log T})$ where $\tilde{V}'_T = \sum_{t=1}^{T} \left\| \frac{r_t \circ w_t}{r_t^\top w_t} - \frac{r_{t-1} \circ w_t}{r_{t-1}^\top w_t} \right\|_2^2$ by using OUE-LB-FTRL. We set the current prediction $D_t$ as a delta distribution on $r_{t-1}$ and using the logarithmic utility function. Since $\tilde{V}'_T \leq 2T$, we have a $O(\sqrt{nT \log T})$ worst-case regret bound. Table 4 summarizes the results on gradual-varioation bounds.

## 2.5. Online Convex Optimization with Predicted Functions

We develop a general regret inequality for the optimistic FTRL algorithm with convex hint functions. We use our result to obtain the regret bounds of all the algorithms in this paper. Our analysis

Table 4: Gradual-Variation Regret Bounds for Online Portfolio Selection

| Algorithm | First-Order Regret | Worst-case Regret | Run-time |
|---|---|---|---|
| UOCO (Yan et al., 2023) | $Rn \log V_T$ | $\infty$ | $n^3 \log T$ |
| Tsai et al. (2023b) | $\sqrt{n \tilde{V}_T \log T}$ | $\sqrt{nT} \log T$ | $n$ |
| OUE-LB-FTRL (Theorem 9) | $\sqrt{n \tilde{V}'_T \log T}$ | $\sqrt{nT \log T}$ | $n^3$ |

is a novel extension over the current literature which only considers linear hint functions. These results and prior works appear in Appendix A.

## 3. AdaCurv ONS

We first show a new adaptive curvature quadratic surrogate function for $-\log(r_t^\top w)$.

**Lemma 1** *For all $x, y \in \Delta_n, r_t \in \mathbb{R}_+^n$ such that $r_t^\top x, r_t^\top y > 0$, we have the inequality:*

$$-\log(r_t^\top x) \geq -\log(r_t^\top y) - \frac{r_t^\top (x - y)}{r_t^\top y} + \frac{r_t^\top y}{2 \max_i r_t(i)} \left( \frac{r_t^\top (x - y)}{r_t^\top y} \right)^2$$

Lemma 1 implies the following surrogate function for $f_t(w) = -\log(r_t^\top w)$:

$$\tilde{f}_t(w) = f_t(w_t) + \nabla f_t(w_t)^\top (w - w_t) + \frac{r_t^\top w_t}{2(\max_i r_t(i))} (\nabla f_t(w_t)^\top (w - w_t))^2 \qquad (1)$$

Plugging the surrogate function into the ONS algorithm of Hazan et al. (2007), we get the Adaptive Curvature ONS (AdaCurv ONS) update:

$$w_t \in \arg \min_{w \in \Delta_n} \sum_{s=1}^{t-1} \tilde{f}_s(w) + \frac{\epsilon}{2} \|w\|_2^2 \qquad (2)$$

AdaCurv FTAL is obtained by using $\epsilon = 0$ in AdaCurv ONS. AdaCurv FTAL has no parameters to tune and the iterates $w_t$ are invariant to scaling of returns as shown below:

$$w_t \in \arg \min_{w \in \Delta_n} \left( \sum_{s=1}^{t-1} -\frac{r_s^\top w}{r_s^\top w_s} + \sum_{s=1}^{t-1} -\frac{r_s^\top w}{\max_i r_s(i)} \right) + \frac{1}{2} \left( \sum_{s=1}^{t-1} \frac{(r_s^\top w)^2}{(r_s^\top w_s)(\max_i r_s(i))} \right)$$

### 3.1. AdaCurv ONS Regret Bound

**Theorem 2** *For $w \in \Delta$, any sequence of returns $r_1, \ldots, r_T \in \mathbb{R}_+^n$, define $f_t(w) = -\log(r_t^\top w)$. With $\epsilon = 1$, AdaCurv ONS (Equation (2)) has the data-dependent regret bound:*

$$\sum_{t=1}^T f_t(w_t) - f_t(w) \leq \frac{1}{2} + \frac{nR}{2} \log(1 + TR)$$

*If we set $\epsilon = 0$, we get the data-dependent regret bound for AdaCurv FTAL:*

$$\sum_{t=1}^T f_t(w_t) - f_t(w) \leq R + \frac{nR}{2} \log(1 + T^2)$$

If the returns $r_t \in [c, C]^n$, then $R = C/c$. The regret of AdaCurv ONS for any sequence of such returns is bounded by $O((C/c)n \log T)$. However, AdaCurv ONS does not need to know the values of $C$ and $c$ beforehand to achieve this regret. Whereas the ONS algorithm of Hazan et al. (2007) would need to known $C$ and $c$ to achieve the same regret bound.

Consider a sequence of returns where $\min_i r_t(i) = 1/t$ and $\max_i r_t(i) = 1$. For such a sequence, $R = T$. Thus, the regret of AdaCurv ONS may be unbounded for such sequences.

## 4. LB-AdaCurv ONS

While AdaCurv ONS has $O(nR \log T)$ data-dependent regret, it's worst case regret is not uniformly bounded for all sequences of returns. Indeed, $R$ could be $O(T)$ for some sequences of returns. We address this issue by using the log-barrier regularizer. By tuning the strength of the regularization, we are able to obtain a worst-case regret bound while maintaining the data-dependent regret bound. We consider updates of the form:

$$w_t \in \arg \min_{w \in \Delta_n} \sum_{s=1}^{t-1} \tilde{f}_s(w) + \frac{\epsilon}{2} \|w\|_2^2 + \frac{1}{\eta_{t-1}} F(w) \tag{3}$$

Here $F(w) = \sum_{i=1}^{n} [\log(1/n) - \log(w(i))]$, is the log-barrier regularizer, $\tilde{f}_t(w)$ are adaptive curvature surrogate functions of the form in Equation (1).

The LB-AdaCurv ONS algorithm is described in Algorithm 1. The BARRONS algorithm of Luo et al. (2018) also utilizes a log-barrier regularized ONS procedure. However, their algorithm uses the a constant-curvature surrogate function and an increasing sequence of parameters $\eta_t$. On the other hand, we use an adaptive surrogate function and a decreasing sequence of parameters $\eta_t$ chosen via the AdaFTRL (Orabona and Pál, 2018) technique.

**Theorem 3** *For $w \in \Delta$, any sequence of returns $r_1, \ldots, r_T \in \mathbb{R}_+^n$, define $f_t(w) = -\log(r_t^\top w)$. If we set $\epsilon = 1$, we get the bound for LB-AdaCurv ONS:*

$$\sum_{t=1}^{T} f_t(w_t) - f_t(w) \leq \frac{5}{2} + 2n \log T + \min \left( nR \log (1 + RT), 2 + 2\sqrt{2nT \log(T)} \right)$$

*If we set $\epsilon = 0$, we get the bound for LB-AdaCurv FTAL:*

$$\sum_{t=1}^{T} f_t(w_t) - f_t(w) \leq 2 + 2n \log(T) + \min \left( 2R + nR \log (1 + T^2), 2 + 2\sqrt{2nT \log(T)} \right)$$

LB-AdaCurv ONS/FTAL maintains the $O(nR \log T)$ data-dependent regret of AdaCurv ONS/FTAL while guaranteeing a worst-case regret of $O(\sqrt{nT \log T})$.

## 5. Optimistic Expected Utility LB-FTRL

Consider an investor who at time $t$ has a prediction distribution $D_t$ and a utility function $U$. We augment the log-barrier regularized FTRL algorithm with the expected utility of the player by using the following update:

$$w_t \in \arg \min_{w \in \Delta_n} \sum_{s=1}^{t-1} \nabla f_t(w_t)^\top w + \frac{F(w)}{\eta_{t-1}} - \mathbb{E}_{r \sim D_t}[U(r^\top w)]$$

---

**Algorithm 1:** Log-Barrier Regularized Adaptive Curvature Online Newton Step

---

Input Parameter: $\epsilon$

Starting Parameters: $\eta_0 = 1/2$

Regularizer $F(q) = \sum_{i=1}^{n}(f(w(i)) - f(1/n))$, where $f(x) = -\log(x)$

**for** $t = 1$ *to* $T$ **do**

    Pick portfolio: $w_t \in \arg\min_{w \in \Delta_n} \sum_{s=1}^{t-1} \tilde{f}_s(w) + \frac{\epsilon}{2}\|w\|_2^2 + \frac{1}{\eta_{t-1}}F(w)$

    Observe returns vector $r_t$. Let $f_t(w) = -\log(r_t^\top w)$

    Construct adaptive curvature surrogate function

$$\tilde{f}_t(w) = f_t(w_t) + \nabla f_t(w_t)^\top (w - w_t) + \frac{r_t^\top w_t}{2(\max_i r_t(i))}(\nabla f_t(w_t)^\top (w - w_t))^2$$

    Let $\tilde{g}_t = \sum_{s=1}^{t} \tilde{f}_t$. Compute $M_t(\eta_{t-1}) =$

$$\sup_{w \in \Delta_n} \left[ \nabla f_t(w_t)^\top (w_t - w) - \mathbf{B}_{\tilde{g}_t}(w\|w_t) - \frac{\epsilon}{2}\|w - w_t\|_2^2 - \frac{1}{\eta_{t-1}}\mathbf{B}_F(w\|w_t) \right]$$

    Compute $\eta_t = \dfrac{n \log T}{2n \log T + \sum_{s=1}^{t} M_s(\eta_{s-1})}$

**end**

---

Here $F(w) = \sum_{i=1}^{n}[\log(1/n) - \log(w(i))]$ is the log-barrier regularizer. Let $U'$ is the first derivative of $U$ and $C$ is a constant chosen such that $C = 1 + \sup_x xU'(x)$. The OEU-LB-FTRL algorithm is described in Algorithm 2.

---

**Algorithm 2:** Optimistic Expected Utility Log-Barrier FTRL (OEU-LB-FTRL)

---

Starting Parameters: $\eta_0 = 1/2$

Pick $C = 1 + \sup_x xU'(x)$

Regularizer $F(q) = \sum_{i=1}^{n}(f(w(i)) - f(1/n))$, where $f(x) = -\log(x)$

**for** $t = 1$ *to* $T$ **do**

    Investor has a prediction distribution $D_t$ and utility function $U_t$

    Pick portfolio: $w_t \in \arg\min_{w \in \Delta_n} \sum_{s=1}^{t-1} \nabla f_t(w_t)^\top w + \frac{F(w)}{\eta_{t-1}} - \mathbb{E}_{r \sim D_t}[U(r^\top w)]$

    Observe returns vector $r_t$. Let $f_t(w) = -\log(r_t^\top w)$

    Let $\tilde{g}_t = \sum_{s=1}^{t} \tilde{f}_t$. Compute :

$$M_t(\eta_{t-1}) = \sup_{w \in \Delta_n} \left[ \left( \nabla f_t(w_t) + \mathbb{E}_{r \sim D_t}[U'(r^\top w_t)r] \right)^\top (w_t - w) - \frac{1}{\eta}\mathbf{B}_F(w\|w_t) \right]$$

    Compute $\eta_t = \frac{n \log T}{Cn \log T + \sum_{s=1}^{t} M_s(\eta_{s-1})}$

**end**

---

**Theorem 4** *For $w \in \Delta$, any sequence of returns $r_1, \ldots, r_T \in \mathbb{R}_+^n$, return prediction distributions $D_1, \ldots, D_T$, concave and strictly increasing utility function $U$ with a strictly decreasing first derivative $U'$, define $f_t(w) = -\log(r_t^\top w)$. The updates of OEU-LB-FTRL (Algorithm 2) satisfy the regret bound:*

$$\sum_{t=1}^T f_t(w_t) - f_t(w) \leq 2 + C\left(1 + 2n\log T\right) + 2\sqrt{2n\left(\sum_{t=1}^T \left\|\mathbb{E}_{r\sim D_t}[U'(r^\top w_t)r \circ w_t] - \frac{r_t \circ w_t}{r_t^\top w_t}\right\|_2^2\right)\log T}$$

*Where $C = 1 + \sup_x xU'(x)$. This implies the worst-case regret bound:*

$$\sum_{t=1}^T f_t(w_t) - f_t(w) \leq 2 + C\left(1 + 2n\log T\right) + 2C\sqrt{2nT\log T}$$

*Moreover, if $U$ and $D_t$ are such that $\mathbb{E}_{r\sim D_t}[U'(r^\top w_t)r \circ w_t] = \frac{r_t \circ w_t}{r_t^\top w_t}$, then we have the regret bound:*

$$\sum_{t=1}^T f_t(w_t) - f_t(w) \leq 2 + 2Cn\log T$$

For the specific case of a Kelly Criteria investor, i.e., $U(x) = \log(x)$, we have $C = 1 + \sup_x xU'(x) = 2$. Thus, the worst case regret of OEU-LB-FTRL with Kelley Criteria is $4 + 4n\log T + 4\sqrt{nT\log T}$. As this regret bound of $O(\sqrt{nT\log T})$ holds for any prediction distribution $D_t$, it constitutes a robustness guarantee. In the scenario where $D_t$ is a delta distribution on $r_t$, we have $\mathbb{E}_{r\sim D_t}\left[U'(r^\top w_t)r \circ w_t\right] = \frac{r_t \circ w_t}{r_t^\top w_t}$, so the regret is $2 + 4n\log T$. Thus, Algorithm 2 with $U(x) = \log(x)$ is $O(n\log T)$-consistent and $O(\sqrt{nT\log T})$-robust with respect to static regret.

## 5.1. Robustness and Consistency

When viewed from the perspective of wealth generated, the result of Theorem 4 is unsatisfactory. Denote by $W(Alg)$ the wealth of the investor and $W(w^\star)$ the wealth of the best static allocation vector $w^\star \in \arg\min_{w\in\Delta_n} \sum_{t=1}^T f_t(w)$. Then, the robustness guarantee ensures that $W(Alg) \geq W(w^\star)\exp(-O(\sqrt{nT\log T}))$. If the prediction distribution $D_t$ perfectly predicts $r_t$, then the consistency guarantee ensures $W(Alg) \geq W(w^\star)\exp(-O(n\log T))$. Considering the fact that a purely regret minimizing investor who uses Cover's UP algorithm can ensure $W(Cover) \geq W(w^\star)\exp(-O(n\log T))$, without even taking the predictions $D_t$ into account, we can see that the consistency guarantee of $W(Alg) \geq W(w^\star)\exp(-O(n\log T))$ with perfect predictions is quite weak. Additionally, with perfect predictions, the wealth of the expected utility player $W(EU)$ will be $\prod_{t=1}^T(\max_i r_t(i))$, which could be exponentially larger than the wealth of the best static allocation $W(w^\star) = \prod_{t=1}^T(r_t^\top w^\star)$.

Ideally, we would like to seek an algorithm whose robustness guarantee ensures that the investor's wealth is close to $W(w^\star)$ in case the returns $r_t$ are completely arbitrary. Additionally, if the predictions are perfect, then the consistency guarantee must ensure that investor's wealth is close to $W(EU)$. In the next section, we present an algorithm that achieves this robustness-consistency guarantee. As the wealth achieved will always track the better of $W(EU)$ or $W(w^\star)$ depending on the accuracy of predictions, our algorithm obtains the best of both worlds.

## 6. Best of Both Worlds for Online Portfolio Selection

Consider an expected utility investor (EU) who picks a portfolio $w_t^{EU}$ at time $t$ by solving the stochastic optimization problem: $w_t^{EU} = \arg\max_{w \in \Delta_n} \mathbb{E}_{r \sim D_t}\left[U(r^\top w)\right]$. Also consider a purely regret minimizing investor (RM) who picks a portfolio $w_t^{RM}$ in round $t$ by using a regret minimizing algorithm like the ones in Table 5. Denote the algorithm used by this player as $RM$. Finally consider an meta-investor who can only allocate wealth to EU and RM. The meta-investor cannot directly participate in the actual market of $n$ assets, but can indirectly participate via the two base investors, EU and RM. At time $t$, the meta-investor allocates $\gamma_t$ portion of wealth to EU and $1 - \gamma_t$ portion to RM. The quantity $\gamma_t$ itself could be chosen by using a regret minimizing algorithm from Table 5. Denote the algorithm used by the meta-investor as $metaRM$. The returns seen by meta-investor are the returns of the base investors, $r_t^{EU} = r_t^\top w_t^{EU}$ and $r_t^{RM} = r_t^\top w_t^{RM}$. So, if the allocation chosen by $metaRM$ is $\gamma_t$, then the implicit allocation of the meta-investor is $\gamma_t w_t^{EU} + (1 - \gamma_t) w_t^{RM}$. The Best of Both Worlds OPS algorithm picks $w_t = \gamma_t w_t^{EU} + (1 - \gamma_t) w_t^{RM}$.

---

**Algorithm 3:** Best of Both Worlds for Online Portfolio Selection (BoB-OPS)

---

**for** $t = 1$ *to* $T$ **do**

    EU investor picks portfolio: $w_t^{EU} = \arg\max\limits_{w \in \Delta_n} \mathbb{E}_{r \sim D_t}\left[U(r^\top w)\right]$

    RM investor picks portfolio: $w_t^{RM} = RM(\{w_1^{RM}, r_1, \dots, w_{t-1}^{RM}, r_{t-1}\})$

    Pick EU-RM allocation $\gamma_t = metaRM(\{\gamma_1, (r_1^{EU}, r_1^{RM}), \dots, \gamma_{t-1}, (r_{t-1}^{EU}, r_{t-1}^{RM})\})$

    Pick portfolio: $w_t = \gamma_t w_t^{EU} + (1 - \gamma_t) w_t^{RM}$

    See returns $r_t$. Set $r_t^{EU} = r_t^\top w_t^{EU}$ and $r_t^{RM} = r_t^\top w_t^{RM}$

**end**

---

**Theorem 5** *For $w \in \Delta_n$, any sequence of returns $r_1, \dots, r_T \in \mathbb{R}_+^n$, return prediction distributions $D_1, \dots, D_T$, concave and strictly increasing utility function $U$ with a strictly decreasing first derivative $U'$, define $f_t(w) = -\log(r_t^\top w)$. The updates of BoB-OPS (Algorithm 3) satisfy the regret bounds:*

$$\sum_{t=1}^{T} f_t(w_t) - f_t(w_t^{EU}) \le \mathcal{R}_{metaRM}(2, T) \quad and \quad \sum_{t=1}^{T} f_t(w_t) - f_t(w) \le \mathcal{R}_{RM}(n, T) + \mathcal{R}_{metaRM}(2, T)$$

*$\mathcal{R}_{RM}(n, T)$ and $\mathcal{R}_{metaRM}(2, T)$ are the regret bounds for the algorithm used by the RM investor and the meta-investor respectively. If the RM investor and meta-RM investor use Cover's Universal Portfolio (Cover, 1991) algorithm satisfy the regret bounds:*

$$\sum_{t=1}^{T} f_t(w_t) - f_t(w_t^{EU}) \le \log(T + 1) \quad and \quad \sum_{t=1}^{T} f_t(w_t) - f_t(w) \le n\log(T + 1)$$

The final part of Theorem 5 implies the following wealth lower bound:

$$W(\text{Bob-OPS}) \ge \max\left(\frac{W(EU)}{T + 1}, \frac{W(w^\star)}{(T + 1)^n}\right)$$

Thus, when the predictions $D_t$ are perfect, we have the consistency guarantee $W(\text{Bob-OPS}) \geq \frac{W(EU)}{T+1}$. When the predictions are arbitrarily bad, we have the robustness guarantee $W(\text{Bob-OPS}) \geq \frac{W(w^\star)}{(T+1)^n}$, where $w^\star \in \arg\min_{w \in \Delta_n} \sum_{t=1}^T f_t(w)$ is the optimal static allocation in hindsight.

## 7. More Data-Dependent Regret Bounds

### 7.1. First-Order Regret Bound

**Theorem 6** *For $w \in \Delta$, any sequence of returns $r_1, \ldots, r_T \in \mathbb{R}_+^n$, define $f_t(w) = -\log(r_t^\top w)$. The updates of AdaCurv ONS (Equation (2)) with $\epsilon = 1$ satisfy the regret bound:*

$$\sum_{t=1}^T f_t(w_t) - f_t(w) \leq \frac{1}{2} + \frac{nR}{2}\log\left(4nR^3\log\left(\frac{4nR^3}{e}\right) + 4R^2 + 8R^2 L_T^\star + 2\right)$$

*Here, $L_T^\star = \min_{w \in \Delta_n}\left[\sum_{t=1}^T f_t(w)\right] - \sum_{t=1}^T\left[\min_{w \in \Delta_n} f_t(w)\right]$ is the regret between the best static and the best dynamic portfolio selection strategies.*

**Theorem 7** *For $w \in \Delta$, any sequence of returns $r_1, \ldots, r_T \in \mathbb{R}_+^n$, define $f_t(w) = -\log(r_t^\top w)$. The updates of LB-AdaCurv ONS (Algorithm 1) with $\epsilon = 1$ satisfy the regret bound:*

$$\sum_{t=1}^T f_t(w_t) - f_t(w_t) \leq \frac{5}{2} + 2n\log T + \min\left[2 + 2\sqrt{8n\log T} + 4\sqrt{8n\left(L_T^\star + \frac{9}{2} + 2n\log T\right)\log T},\right.$$

$$\left. nR\log\left(8nR^3\log\left(\frac{8nR^3}{e}\right) + 20R^2 + 16R^2 n\log T + 8R^2 L_T^\star + 2\right)\right]$$

*Here, $L_T^\star = \min_{w \in \Delta_n}\left[\sum_{t=1}^T f_t(w)\right] - \sum_{t=1}^T\left[\min_{w \in \Delta_n} f_t(w)\right]$ is the regret between the best static and the best dynamic portfolio selection strategies.*

### 7.2. Second-Order Regret Bound

**Theorem 8** *For $w \in \Delta$, any sequence of returns $r_1, \ldots, r_T \in \mathbb{R}_+^n$, define $f_t(w) = -\log(r_t^\top w)$. The AdaCurv ONS updates (Equation (2)) with $\epsilon = 1$ satisfy the regret bound:*

$$\sum_{t=1}^T f_t(w_t) - f_t(w) = O\left(nR^2\log(1 + Q_T + n) + \sqrt{n}R\log(1 + Q_T/n) + 1\right)$$

*Here $Q_T = \min_\mu \sum_{t=1}^T \|r_t - \mu\|_2^2 = \sum_{t=1}^T \|r_t - \bar{r}_T\|_2^2$, where $\bar{r}_T = \frac{1}{T}\sum_{t=1}^T r_t$.*

### 7.3. Gradual-Variation Bound

**Theorem 9** *For $w \in \Delta$, any sequence of returns $r_1, \ldots, r_T \in \mathbb{R}_+^n$, let the return prediction distribution $D_t$ be the delta distribution on $r_{t-1}$ (Let $r_0$ be the all 1s vector). The updates of OEU-LB-FTRL (Algorithm 2) with $U(x) = \log(x)$ satisfy the regret bound:*

$$\sum_{t=1}^T f_t(w_t) - f_t(w) \leq 4 + 4n\log T + 2\sqrt{2n\tilde{V}_T'\log T}$$

*Here $\tilde{V}_T' = \sum_{t=1}^T \left\|\frac{r_t \circ w_t}{r_t^\top w_t} - \frac{r_{t-1} \circ w_t}{r_{t-1}^\top w_t}\right\|_2^2$ and $r_0$ is the all ones vector.*

## 8. Conclusion

In this paper, we first studied data-dependent regret bounds for the Online Portfolio Selection problem. We first obtained a new adaptive curvature surrogate function for the loss $-\log(r_t^\top w)$. We presented two algorithms that use this surrogate function. First is the AdaCurv ONS algorithm that has a data-dependent regret bound of $O(nR \log T)$, where $R = \max_{t,i,j} r_t(i)/r_t(j)$. Next, we proposed the LB-AdaCurv ONS algorithm, which adds the log-barrier regularization to AdaCurv ONS, giving it a regret of $O(\min(nR \log T, \sqrt{nT \log T}))$. The ONS algorithm Hazan et al. (2007) with the constant curvature surrogate functions is an important building block in recent OPS algorithms such as AdaBARRONS Luo et al. (2018), BISONS Zimmert et al. (2022) and PAE+DONS Mhammedi and Rakhlin (2022). As future work, one could explore if the usage of our adaptive curvature surrogate function along with the techniques developed in Luo et al. (2018); Zimmert et al. (2022); Mhammedi and Rakhlin (2022) could lead to more elegant algorithms as it avoids the need to tune the curvature in the quadratic surrogate.

Next, we explored the integration of predicted returns into the online portfolio selection (OPS) framework, presenting a novel approach that bridges the gap between regret minimization techniques and expected utility theory to guide investment decisions. The OEU-LB-FTRL algorithm with logarithmic utility is shown to be $O(n \log T)$-consistent and $O(\sqrt{nT \log T})$-robust with respect to static regret. We improve these results using the BoB-OPS algorithm and show that it has $O(\log T)$ regret with respect to the expected utility investor and $O(n \log T)$ static regret.

Finally, we showed more data dependent regret bounds for our algorithms. We showed that AdaCurv ONS has a logarithmic first-order regret bound of $O(nR \log L_T^\star)$ and LB-AdaCurv ONS has a first-order regret bound of $O(\min(nR \log L_T^\star, \sqrt{nL_T^\star \log T}))$. AdaCurv ONS is also shown to obtain a second-order regret bound of $O(nR^2 \log Q_T)$. However, $R$ could be unbounded. Finding an algorithm achieving a logarithmic second order regret bound in $Q_T$, while also having a bounded worst case regret remains open problem. The OEU-LB-FTRL algorithm with log utility and previous return as the prediction gives a gradual variation bound of $O(\sqrt{n\tilde{V}_T' \log T})$. Similarly, finding an algorithm with logarithmic gradual variation measured in $V_T$ and bounded worst case regret remains open problem.

The general regret inequality developed in Appendix A for Optimistic FTRL with convex hint functions provides a novel extension over the current literature, which only consider linear hint functions. These general regret bound could be of independent interest to the OCO research community.

### Acknowledgments

This research was supported by NSF CAREER award number 1846792: Consumer Behavior-Aware Learning for Revenue Management.

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

Table 5: Worst-case Regret Bounds for Online Portfolio Selection

| Algorithm | Worst-case Regret | Run-time | Returns Domain |
|---|---|---|---|
| UP (Cover, 1991; Kalai and Vempala, 2002) | $n \log(T)$ | $n^4 T^{14}$ | $\mathbb{R}_+^n$ |
| EG (Helmbold et al., 1998) | $(C/c)\sqrt{T \log n}$ | $n$ | $[c, C]^n, 0 < c < C$ |
| EG + Universalization (Helmbold et al., 1998; Tsai et al., 2023a) | $n^{1/3} T^{2/3}$ | $n$ | $\mathbb{R}_+^n$ |
| ONS(Hazan et al., 2007) | $(C/c)n \log T$ | $n^3$ | $[c, C]^n, 0 < c < C$ |
| ONS + Universalization | $n\sqrt{T \log T}$ | $n^3$ | $\mathbb{R}_+^n$ |
| Soft-Bayes (Orseau et al., 2017) | $\sqrt{nT \log n}$ | $n$ | $\mathbb{R}_+^n$ |
| AdaBARRONS (Luo et al., 2018) | $n^2 \log^4 T$ | $n^{2.5}T$ | $\mathbb{R}_+^n$ |
| BISONS (Zimmert et al., 2022) | $n^2 \log^2 T$ | $n^3$ | $\mathbb{R}_+^n$ |
| PAE+DONS (Mhammedi and Rakhlin, 2022) | $n^2 \log^5 T$ | $n^3$ | $\mathbb{R}_+^n$ |
| LB-OMD(Tsai et al., 2023a) | $\sqrt{nT \log T}$ | $n$ | $\mathbb{R}_+^n$ |
| VB-FTRL(Jézéquel et al., 2022) | $n \log T$ | $n^2 T$ | $\mathbb{R}_+^n$ |

## Appendix A. Online Convex Optimization with Predicted Functions

The Online Convex Optimization framework (OCO) was first defined by Zinkevich (2003). It provides a powerful framework for the design and analysis of regret minimizing algorithms. In the last two decades, there have been many developments in this area and it continues to be an active area of research within the the machine learning, operations research and statistics communities. It has also seen widespread adoption by practitioners. Algorithms that originate from OCO, like AdaGrad Duchi et al. (2011) and Adam Kingma and Ba (2015) are widely used as optimizers for training deep neural networks. The monographs of Hazan (2016); Shalev-Shwartz (2012); Orabona (2019) provide a comprehensive overview of OCO.

A player interacts with an environment for $T$ rounds. In each round, the player selects an action from a convex set, $w_t \in \mathcal{D} \subseteq \mathbb{R}^n$. The environment picks a convex function $f_t : \mathcal{D} \to \mathbb{R}$. The player incurs a scalar cost $f_t(w_t)$ and observes the function $f_t$. The player's objective is to minimize the total cost of interaction over the $T$ rounds $\sum_{t=1}^T f_t(w_t)$. The *static-regret* (regret for short) of the player compared to the cost of fixed point $w \in \mathcal{D}$ is $\sum_{t=1}^T f_t(w_t) - f_t(w)$. The action $w_t$ is selected using an algorithm $\mathcal{A}$, that takes as input the current information set $\mathcal{I}_t = \{w_1, f_1, \ldots, w_{t-1}, f_{t-1}\}$ and outputs the action, i.e. $w_t = \mathcal{A}(I_t)$. The interaction protocol is summarized below:

A straightforward strategy for the player is to select $w_t$ using *Follow The Leader* (FTL). FTL can be succinctly expressed as:

$$w_t \in \arg\min_{w \in \mathcal{D}} \sum_{s=1}^{t-1} f_s(w) \qquad \text{(FTL)}$$

Unfortunately, FTL can have $O(T)$ regret even with linear functions (Orabona, 2019, Example 2.10). This occurs because FTL's iterates can be forced into alternating between opposite corners of $\mathcal{D}$ in every iteration, making it *"unstable"*. Nevertheless, FTL has $O(\log T)$ regret when the functions are strongly convex (Orabona, 2019, Corollary 7.24). Even for linear functions, FTL's

---

**Online Convex Optimization - Interaction Protocol:**

Initial information set $\mathcal{I}_1 = \{\}$

**for** $t = 1$ *to* $T$ **do**

    Player picks $w_t = \mathcal{A}(\mathcal{I}_t)$

    Environment picks $f_t$

    Player incurs cost $f_t(w_t)$

    Update information set $\mathcal{I}_{t+1} = I_t \cup \{w_t, f_t\}$

**end**

---

regret is $O(\log T)$ if the decision set's boundary exhibits sufficient curvature Huang et al. (2016, 2017).

Incorporating regularization into FTL is a common approach to *"stabilize"* the iterations, resulting in a widely studied algorithm in the OCO literature called Follow The Regularized Leader (FTRL). Another popular algorithm for OCO is Online Mirror Descent (OMD), which stabilizes the iterates by ensuring consecutive iterates remain close to each other. Several fascinating connections and equivalences exist between FTRL and OMD, as discussed in Orabona (2019). Various well-known iterative algorithms in machine learning, such as Online Gradient Descent Zinkevich (2003), AdaGrad Duchi et al. (2011), Exponentiated GradientKivinen and Warmuth (1997), and Online Newton Step Hazan et al. (2007), can be formulated using one of these two algorithms. See Orabona (2019) for a detailed history of FTRL. In this paper, we focus on the FTRL algorithm and its variants. In its simplest form, it can be stated as:

$$w_t \in \arg\min_{w \in \mathcal{D}} \sum_{s=1}^{t-1} f_s(w) + \frac{F(w)}{\eta_{t-1}} \tag{FTRL}$$

Here, $F(w)$ is the regularization function and $\eta_{t-1}$ is a time varying learning-rate parameter that needs to be tuned. While there are several techniques for picking $\eta_{t-1}$, we focus on the AdaFTRL technique prescribed by Orabona and Pál (2018) for obtaining data-dependent regret bounds.

The above FTRL requires minimizing the sum of $t$ convex functions in round $t$. In general, this could potentially require $O(t)$ computation per round, becoming increasingly costly as the number of rounds increases. One can construct *surrogate* convex functions that are linear or quadratic and run FTRL on them to mitigate this computational issue. Assume we have a surrogate function $\tilde{f}_t$ such that $f_t(w_t) = \tilde{f}_t(w_t)$ and $f_t(w) \geq \tilde{f}_t(w)$ for all $w \in \mathcal{D}$, then we have:

$$\sum_{t=1}^{T} f_t(w_t) - f_t(w) \leq \sum_{t=1}^{T} \tilde{f}_t(w_t) - \tilde{f}_t(w)$$

Thus, if we can bound the regret of the surrogates $\sum_{t=1}^{T} \tilde{f}_t(w_t) - \tilde{f}_t(w)$ using FTRL on $\tilde{f}_t$, we obtain a bound on the regret $\sum_{t=1}^{T} f_t(w_t) - f_t(w)$. Running an FTRL on $\tilde{f}_t$ instead of $f_t$ not only delivers computational benefits, but may also aid in obtaining tighter regret bounds.

In the online learning with predictions (OLP) framework, the player is given a predicted function $m_t(w)$ before picking $w_t$. The interaction protocol for OLP is below:

For OLP, we employ the Optimistic FTRL Rakhlin and Sridharan (2013a) algorithm:

---

**Online Convex Optimization with Predictions - Interaction Protocol:**

Initial information set $\mathcal{I}_1 = \{\}$
**for** $t = 1$ *to* $T$ **do**
> Receive prediction $m_t$
> Player picks $w_t = \mathcal{A}(\mathcal{I}_t \cup \{m_t\})$
> Environment picks $f_t$
> Player incurs cost $f_t(w_t)$
> Update information set $\mathcal{I}_{t+1} = I_t \cup \{w_t, f_t\}$

**end**

---

$$w_t \in \arg\min_{w \in \mathcal{D}} \sum_{s=1}^{t-1} f_s(w) + m_t(w) + \frac{F(w)}{\eta_{t-1}} \qquad \text{(OFTRL)}$$

The OLP framework, while formally introduced in Rakhlin and Sridharan (2013a,b), had previously appeared in various forms Azoury and Warmuth (2001); Chiang et al. (2012). This framework has been instrumental in demonstrating several intriguing results, such as adaptive regret bounds in online learning Steinhardt and Liang (2014), adaptive regret bounds in adversarial bandits Wei and Luo (2018), and accelerated rates of convergence for two-player games Syrgkanis et al. (2015), to name a few. A related area of research, called *Algorithms with Predictions* Mitzenmacher and Vassilvitskii (2022) studies how predictions could be used to improve the performance of online algorithms, where the performance benchmark is competitive ratio. While the problems studied in this area are different from OLP, they share the common goal of going beyond wort-case performance with the help of predictions.

### A.1. Regret Inequality for Optimistic FTRL with Convex Predictions

In prior works such as Rakhlin and Sridharan (2013a), the regret inequality is obtained for linear costs and linear predictions (See Luo (2022) for a simple proof). We extend their result and obtain a general regret inequality for Optimistic FTRL with convex cost functions $f_t$ and convex predictions $m_t$. Our result is stated in terms of *Bregmen Divergences* and *Mixed-Bregmans*.

**Definition 10 (Bregman Divergence)** *The Bregman Divergence of function $F$ is:*

$$B_F(x\|y) = F(x) - F(y) - \nabla F(y)^\top (x - y)$$

**Definition 11 (Mixed Bregman)** *For $\alpha, \beta > 0$ the $(\alpha, \beta)$-Mixed Bregman of function $F$ is:*

$$B_F^{\alpha,\beta}(x\|y) = \frac{F(x)}{\alpha} - \frac{F(y)}{\beta} - \frac{\nabla F(y)}{\beta}^\top (x - y)$$

The Mixed Bregman is not a divergence as $B_F^{\alpha,\beta}(x\|x)$ may not be zero. However, we do have the relation $\alpha B_F^{\alpha,\alpha}(x\|y) = B_F(x\|y)$.

Let the iterates of Optimistic FTRL be $w_t$:

$$w_t \in \arg\min_{w \in \mathcal{D}} \sum_{s=1}^{t-1} f_s(w) + m_t(w) + \frac{F(w)}{\eta_{t-1}}$$

Let the iterates of FTRL be $w_t'$:

$$w_t' \in \arg\min_{w \in \mathcal{D}} \sum_{s=1}^{t-1} f_s(w) + \frac{F(w)}{\eta_{t-1}}$$

We will use the shorthand $g_t = \sum_{s=1}^{t} f_s$. The following theorem bounds the regret of Optimistic FTRL in terms of the iterates $w_t$ and $w_t'$. The proof appears in Appendix A

**Theorem 12** *For any $w \in \mathcal{D}$, any sequence of convex cost functions $f_1, \ldots, f_T$, convex hint functions $m_1, \ldots, m_T$, convex regularizer $F$ and parameters $\eta_0, \ldots, \eta_T$ such that $w_t \in \arg\min_{w \in \mathcal{D}} \sum_{s=1}^{t-1} f_s(w) + m_t(w) + \frac{F(w)}{\eta_{t-1}}$ and $w_t' \in \arg\min_{w \in \mathcal{D}} \sum_{s=1}^{t-1} f_s(w) + \frac{F(w)}{\eta_{t-1}}$. Let $g_t = \sum_{s=1}^{t} f_s$. The iterates of Optimistic FRTL $w_1, \ldots, w_T$ satisfies the regret inequality $\sum_{t=1}^{T} f_t(w_t) - f_t(w)$:*

$$\leq B_F^{\eta_T, \eta_0}(w \| w_1') + \sum_{t=1}^{T} \left[ (\nabla f_t(w_t) - \nabla m_t(w_t))^\top (w_t - w_{t+1}') - B_{g_t}(w_{t+1}' \| w_t) - B_F^{\eta_t, \eta_{t-1}}(w_{t+1}' \| w_t) \right.$$

$$\left. - B_{g_{t-1}}(w_t \| w_t') - B_F^{\eta_{t-1}, \eta_{t-1}}(w_t \| w_t') \right]$$

*Further, if $F$ is such that $\min_{w \in \mathcal{D}} F(w) = 0$ and the sequence $\eta_0, \ldots, \eta_T$ is non-increasing, then the above bound simplifies to $\sum_{t=1}^{T} f_t(w_t) - f_t(w)$:*

$$\leq \frac{F(w)}{\eta_T} + \sum_{t=1}^{T} \left[ (\nabla f_t(w_t) - \nabla m_t(w_t))^\top (w_t - w_{t+1}') - B_{g_t}(w_{t+1}' \| w_t) - \frac{B_F(w_{t+1}' \| w_t)}{\eta_{t-1}} \right.$$

$$\left. - B_{g_{t-1}}(w_t \| w_t') - \frac{B_F(w_t \| w_t')}{\eta_{t-1}} \right]$$

In most applications, the last two terms in the summation are ignored, giving us the inequality $\sum_{t=1}^{T} f_t(w_t) - f_t(w)$:

$$\leq \frac{F(w)}{\eta_T} + \sum_{t=1}^{T} \left[ (\nabla f_t(w_t) - \nabla m_t(w_t))^\top (w_t - w_{t+1}') - B_{g_t}(w_{t+1}' \| w_t) - \frac{B_F(w_{t+1}' \| w_t)}{\eta_{t-1}} \right] \quad (4)$$

However, these two terms do play a role in certain applications, like in showing convergence in general convex games Farina et al. (2022), data-dependent regret bounds in Multi-Armed Bandits Wei and Luo (2018) and gradual-variation bounds in online learning Chiang et al. (2012). In Orabona (2019), a similar regret bound for Optimistic FTRL is obtained, which when translated into our notation would imply $\sum_{t=1}^{T} f_t(w_t) - f_t(w)$:

$$\leq \frac{F(w)}{\eta_T} + \sum_{t=1}^{T} \left[ (\nabla f_t(w_t) - \nabla m_t(w_t))^\top (w_t - w_{t+1}) - B_{g_t}(w_{t+1} \| w_t) - \frac{B_F(w_{t+1} \| w_t)}{\eta_{t-1}} \right] \quad (5)$$

In our bound, we bound the regret of Optimistic FTRL in terms of iterates of both Optimistic FTRL $w_t$ and FTRL $w'_t$. Whereas in Orabona (2019), only the iterates of Optimistic FTRL appear. In our analysis of Optimistic FTRL, we separate the hint $m_t(w)$ from the regularizer term $F(w)/\eta_{t-1}$. On the other hand, in Orabona (2019), the sum $m_t(w) + F(w)/\eta_{t-1}$ is treated as a composite regularizer and analyzed using their FTRL bound. Our general regret bound in Theorem 12 is novel as it could be useful in applications that require the last two terms, like Farina et al. (2022); Wei and Luo (2018). These two terms cannot be obtained via the analysis in Orabona (2019).

In the case where we have no hints, i.e., $m_t = 0$, Equation (4) and Equation (5) become equivalent to the well known FTRL regret inequality. The iterates of FTRL are given by:

$$w_t \in \arg\min_{w \in \mathcal{D}} \sum_{s=1}^{t-1} f_s(w) + \frac{F(w)}{\eta_{t-1}}$$

The regret of FTRL is stated in the following Corollary.

**Corollary 13** *For any $w \in \mathcal{D}$, any sequence of convex cost functions $f_1, \ldots, f_T$ and parameters $\eta_0, \ldots, \eta_T$ such that $w_t \in \arg\min_{w \in \mathcal{D}} \sum_{s=1}^{t-1} f_s(w) + \frac{F(w)}{\eta_{t-1}}$. Assume $F$ is such that $\min_{w \in \mathcal{D}} F(w) = 0$ Let $g_t = \sum_{s=1}^{t} f_s$. The iterates of FRTL $w_1, \ldots, w_T$ satisfies the regret inequality:*

$$\sum_{t=1}^{T} f_t(w_t) - f_t(w) \leq \frac{F(w)}{\eta_T} + \sum_{t=1}^{T} \left[ \nabla f_t(w_t)^\top (w_t - w_{t+1}) - B_{g_t}(w_{t+1} \| w_t) - \frac{B_F(w_{t+1} \| w_t)}{\eta_{t-1}} \right]$$

**Proof** When $m_t = 0$, the iterates of Optimistic FTRL $w_t$ and FTRL $w'_t$ coincide. So, the last two terms in the result of Theorem 12 vanish, i.e., $B_{g_{t-1}}(w_t \| w'_t) = 0$ and $B_F^{\eta_{t-1}, \eta_{t-1}}(w_t \| w'_t) = 0$. ∎

In some applications, besides the regularizer $F$ whose strength is regulated through $\eta_t$, we may need to add an extra constant regularizer $G$ to the optimization. The update equation here is:

$$w_t \in \arg\min_{w \in \mathcal{D}} \sum_{s=1}^{t-1} f_s(w) + G(w) + \frac{F(w)}{\eta_{t-1}}$$

Note that this is different from the optimistic FTRL update where the hint $m_t$ may change over time. In the above update, $G$ is treated as a regularizer and not as a hint. The regret inequality for this update is:

**Corollary 14** *For any $w \in \mathcal{D}$, any sequence of convex cost functions $f_1, \ldots, f_T$ and parameters $\eta_0, \ldots, \eta_T$ such that $w_t \in \arg\min_{w \in \mathcal{D}} \sum_{s=1}^{t-1} f_s(w) + G(w) + \frac{F(w)}{\eta_{t-1}}$. Assume $G, F$ are such that $\min_{w \in \mathcal{D}} G(w) = 0$ and $\min_{w \in \mathcal{D}} F(w) = 0$. Let $g_t = \sum_{s=1}^{t} f_s$. The iterates $w_1, \ldots, w_T$ satisfies the regret inequality $\sum_{t=1}^{T} f_t(w_t) - f_t(w)$:*

$$\leq G(w) + \frac{F(w)}{\eta_T} + \sum_{t=1}^{T} \left[ \nabla f_t(w_t)^\top (w_t - w_{t+1}) - B_{g_t}(w_{t+1} \| w_t) - B_G(w_{t+1} \| w_t) - \frac{B_F(w_{t+1} \| w_t)}{\eta_{t-1}} \right]$$

**Proof** We can apply Corollary 13 starting from time $t = 0$. We use $f_0(w) = G(w)$, and $\eta_{-1} > 0$ in Corollary 13. This gives the regret inequality:

$$\sum_{t=0}^{T} f_t(w_t) - f_t(w) \leq \frac{F(w)}{\eta_T} + \sum_{t=0}^{T} \left[ \nabla f_t(w_t)^\top (w_t - w_{t+1}) - \mathbf{B}_{g_t}(w_{t+1} \| w_t) - \mathbf{B}_G(w_{t+1} \| w_t) \frac{\mathbf{B}_F(w_{t+1} \| w_t)}{\eta_{t-1}} \right]$$

We can write the left hand side of the above inequality as:

$$\sum_{t=0}^{T} f_t(w_t) - f_t(w) = G(w_0) - G(w) + \sum_{t=1}^{T} f_t(w_t) - f_t(w)$$

On the right hand side, we simply the term inside the sum when $t = 0$ as:

$$\nabla G(w_0)^\top (w_0 - w_1) - \mathbf{B}_G(w_1 \| w_0) - \frac{\mathbf{B}_F(w_1 \| w_0)}{\eta_{-1}} \leq \nabla G(w_0)^\top (w_0 - w_1) - \mathbf{B}_G(w_1 \| w_0)$$

$$= G(w_0) - G(w_1) \leq G(w_0)$$

For $t = 1 \ldots T$, the term inside the sum is:

$$\nabla f_t(w_t)^\top (w_t - w_{t+1}) - \mathbf{B}_{g_t}(w_{t+1} \| w_t) - \mathbf{B}_G(w_{t+1} \| w_t) - \frac{\mathbf{B}_F(w_{t+1} \| w_t)}{\eta_{t-1}}$$

Putting the two sides together and simplifying, we get the stated result. ∎

### A.2. Tuning $\eta_t$ using the AdaFRTL technique

In Theorem 12, Corollary 13 and Corollary 14, the inequalities contain the following common form for suitable of $A, b_t$ and $C_t$.

$$\frac{A}{\eta_T} + \sum_{t=1}^{T} b_t^\top (w_t - w_{t+1}) - \mathbf{B}_{C_t(\eta_{t-1})}(w_{t+1} \| w_t)$$

The AdaFTRL strategy picks a specific sequence of parameters $\eta_{t-1}$ based on the history $\mathcal{I}_t$. This strategy was analyzed in Orabona and Pál (2018) and a simpler analysis was given by Koolen (2016). They give a simple algorithmic technique for tuning $\eta_{t-1}$. Our analysis is adapted from Hadiji and Stoltz (2023). We consider time varying parameters of the form:

$$\eta_t = \frac{\alpha}{\beta + \sum_{s=1}^{t} M_s(\eta_{s-1})}$$

Where $\alpha, \beta > 0$ are constants and $M_t(\eta)$ is the optimal value of the following optimization.

$$M_t(\eta) = \sup_{w \in \mathcal{D}} b_t^\top (w_t - w) - \mathbf{B}_{C_t(\eta)}(w \| w_t)$$

Thus, we have the sum:

$$\frac{A}{\eta_T} + \sum_{t=1}^{T} b_t^\top (w_t - w_{t+1}) - \mathbf{B}_{C_t(\eta_{t-1})}(w_{t+1} \| w_t) \leq \frac{A}{\eta_T} + \sum_{t=1}^{T} M_t(\eta_{t-1})$$

We bound the above sum using the following lemma:

**Lemma 15** *Let $\eta_t = \frac{\alpha}{\beta + \sum_{s=1}^t M_s(\eta_{s-1})}$. If $0 \le M_t(\eta_{t-1}) \le L$ for all $t = 1, \ldots T$ and $\frac{M_t(\eta_{t-1})}{\eta_{t-1}} \le g_t$, then we have the upper bound:*

$$\frac{A}{\eta_T} + \sum_{t=1}^T M_t(\eta_{t-1}) \le A\left(\frac{\beta}{\alpha} + \frac{L}{\alpha}\right) + L + \sqrt{2\sum_{t=1}^T g_t\left(\frac{A}{\sqrt{\alpha}} + \sqrt{\alpha}\right)}$$

**Proof** Substituting for $\eta_T$, we have:

$$\frac{A}{\eta_T} + \sum_{t=1}^T M_t(\eta_{t-1}) = \frac{A\beta}{\alpha} + \left(\frac{A}{\alpha} + 1\right)\sum_{t=1}^T M_t(\eta_{t-1})$$

Consider $\left(\sum_{t=1}^T M_t(\eta_{t-1})\right)^2$

$$\left(\sum_{t=1}^T M_t(\eta_{t-1})\right)^2 = \sum_{t=1}^T M_t(\eta_{t-1})^2 + 2\sum_{t=1}^T M_t(\eta_{t-1})\sum_{s=1}^{t-1} M_s(\eta_{s-1})$$

$$= \sum_{t=1}^T M_t(\eta_{t-1})^2 + 2\sum_{t=1}^T M_t(\eta_{t-1})\left(\frac{\alpha}{\eta_{t-1}} - \beta\right)$$

$$\le \sum_{t=1}^T M_t(\eta_{t-1})^2 + 2\alpha\sum_{t=1}^T \frac{M_t(\eta_{t-1})}{\eta_{t-1}}$$

$$\le L\sum_{t=1}^T M_t(\eta_{t-1}) + 2\alpha\sum_{t=1}^T g_t$$

Using the fact that $x^2 \le a + bx$ implies that $x \le \sqrt{a} + b$ for all $a, b, x \ge 0$, we have:

$$\sum_{t=1}^T M_t(\eta_{t-1}) \le \sqrt{2\alpha\sum_{t=1}^T g_t} + L$$

Thus, we get:

$$\frac{A\beta}{\alpha} + \left(\frac{A}{\alpha} + 1\right)\sum_{t=1}^T M_t(\eta_{t-1}) \le \frac{A\beta}{\alpha} + \left(\frac{A}{\alpha} + 1\right)\left(\sqrt{2\alpha\sum_{t=1}^T g_t} + L\right)$$

$$= A\left(\frac{\beta}{\alpha} + \frac{L}{\alpha}\right) + L + \sqrt{2\sum_{t=1}^T g_t\left(\frac{A}{\sqrt{\alpha}} + \sqrt{\alpha}\right)}$$

■

The constants $\alpha$ and $\beta$ are tuning based on $A$ and $L$ in order to obtain a concrete bound.

### A.3. Logarithmic-Barrier Regularizer

The final piece is the regularizer. In this paper, the regularizer we use is the *Logarithmic-Barrier*. Let $\Delta_n$ be the probability simplex $\{x \in \mathbb{R}^n : \sum_{i=1}^n x(i) = 1, x(i) \geq 0, i \in [n]\}$. The log-barrier regularizer defined on $\Delta_n$ is given by the function:

$$F(x) = -n \log(n) - \sum_{i=1}^n \log(x(i))$$

We explore a few important properties of the log-barrier here.

**Definition 16 (Legendre function)** *A continuous function $F : \mathcal{D} \to \mathbb{R}$ is Legendre if $F$ is strictly convex, continuously differentiable on Interior$(\mathcal{D})$ and $\lim_{x \to \mathcal{D}/Interior(\mathcal{D})} \|\nabla F(x)\| = +\infty$.*

It is easy to verify that the log-barrier is a Legendre function on the domain $\mathbb{R}_+^n$.

A crucial step in the analysis of FTRL involves bounding the so-called stability term $\Psi_x(l)$, which is defined as:

$$\Psi_x(l) = \sup_{y \in \Delta_n} l^\top (x - y) - \mathbf{B}_F(y\|x)$$

Let $y^\star$ be the point in $\Delta_n$ achieving the supremum in the definition of $\Psi_x(l)$. As $F$ is Legendre, the supremum is always attained at a unique $y^\star$ in $\Delta_n$.

Let $H(x)$ be a positive definite matrix for every $x \in \Delta_n$. Define the norm $\|z\|_{H(x)}^2 = z^\top H(x)z$. We call such norms as *local norms*. Let $\omega$ is a non-negative convex function. Suppose a lower bound of the following form holds for all $x, y \in \Delta_n$:

$$\mathbf{B}_F(y\|x) \geq \omega(\|x - y\|_{H(x)})$$

Then, we obtain the following upper-bound for $\Psi_x(l)$:

$$\Psi_x(l) = l^\top (x - y^\star) - \mathbf{B}_F(y^\star\|x) \leq \|l\|_{H(x)^{-1}} \|x - y^\star\|_{H(x)} - \omega(\|x - y^\star\|_{H(x)}) \leq \omega^\star(\|l\|_{H(x)^{-1}})$$

Here $\omega^\star$ is the Fenchel-dual of $\omega$, given by $\omega^\star(t) = \sup_s (st - \omega(s))$.

Using the theory of *self-concordant functions* Nesterov (2018), it is possible to obtain one such bound for $\Psi_x(l)$.

**Definition 17 (Self-Concordant Function)** *A continuous function $F : \mathcal{D} \to \mathbb{R}$ is M self-concordant if $F$ is a Legendre function on $\mathcal{D}$ and satisfies:*

$$|\nabla^3 F(x)[u, u, u]| \leq 2M(\nabla^2 F(x)[u, u])^{3/2} \quad \forall x \in \mathcal{D}, u \in \mathbb{R}^n$$

It is easy to verify that the log-barrier satisfies the self-concordance condition with $M = 1$.

Nesterov (2018) obtains the following lower bound for $\mathbf{B}_F(y\|x)$.

**Lemma 18 (Theorem 5.1.8, Nesterov (2018))** *For any $x, y \in \mathbb{R}_+^n$ and $F(x) = -n \log(n) - \sum_{i=1}^n \log(x(i))$, we have:*

$$\mathbf{B}_F(y\|x) \geq \omega(\|x - y\|_{\nabla^2 F(x)})$$

*Here $\omega(t) = t - \log(1 + t)$.*

Using Lemma 18, we have the following theorem bounding $\Psi_x(l)$.

**Lemma 19** *Let $F(x) = -n\log(n) - \sum_{i=1}^{n} \log(x(i))$. For all $x, y \in \Delta_n$ and $l \in \mathbb{R}^n$ such that $\|l\|_{\nabla^2 F(x)^{-1}} \leq 1$, we have the upper-bound:*

$$l^\top(x-y) - B_F(y\|x) \leq \omega^\star(\|l\|_{\nabla^2 F(x)^{-1}})$$

*where $\omega^\star(t) = -t - \log(1-t)$. Further, if we have $\|l\|_{\nabla^2 F(x)^{-1}} \leq \frac{1}{2}$, then we have:*

$$l^\top(x-y) - B_F(y\|x) \leq \|l\|_{\nabla^2 F(x)^{-1}}^2 = \sum_{i=1}^{n} x(i)^2 l(i)^2$$

**Proof** Using Holder's inequality and Lemma 18, we have:

$$l^\top(x-y) - \mathbf{B}_F(y\|x) \leq \|l\|_{\nabla^2 F(x)^{-1}} \|x-y\|_{\nabla^2 F(x)} - \omega(\|x-y\|_{\nabla^2 F(x)}) \leq \omega^\star(\|l\|_{\nabla^2 F(x)^{-1}})$$

Here $\omega^\star(t) = -t - \log(1-t)$ is the Fenchel-dual of $\omega$. Since the domain of $\omega^\star$ is $(-\infty, 1)$, the above inequality holds when $\|l\|_{\nabla^2 F(x)^{-1}} < 1$. Observe that when $t \in [0, 1/2]$, $\omega^\star(t) \leq t^2$. So, when $\|l\|_{\nabla^2 F(x)^{-1}} \leq 1/2$, we have the bound:

$$l^\top(x-y) - \mathbf{B}_F(y\|x) \leq \|l\|_{\nabla^2 F(x)^{-1}}^2 = \sum_{i=1}^{n} x(i)^2 l(i)^2$$

∎

In applications within online learning, Lemma 19 is typically used when $\|l\|_{\nabla^2 F(x)^{-1}} \leq \frac{1}{2}$ holds.

**Corollary 20** *(Putta and Agrawal, 2022, Corollary 7) Let $f(x) = -\log(x) + C$. For $x, y \in (0, 1]$, we have the lower-bound*

$$B_f(y\|x) = \frac{y}{x} - 1 - \ln\left(\frac{y}{x}\right) \geq \frac{1}{2}\frac{(x-y)^2}{x}$$

Using Corollary 20, we can construct the following new upper-bound:

**Lemma 21** *Let $F(x) = -n\log(n) - \sum_{i=1}^{n} \log(x(i))$. For all $x, y \in \Delta_n$ and $l \in \mathbb{R}^n$ we have the upper-bound:*

$$l^\top(x-y) - B_F(y\|x) \leq \frac{1}{2}\sum_{i=1}^{n} x(i)l(i)^2$$

**Proof** Let $f(x) = -\log(x)$, then $F(x) = \sum_{i=1}^{n} f(x(i)) - f(1/n)$

$$l^\top(x-y) - \mathbf{B}_F(y\|x) = \sum_{i=1}^{n} l(i)(x(i) - y(i)) - \mathbf{B}_f(y(i)\|x(i))$$

Apply Corollary 20

$$\leq \sum_{i=1}^{n} l(i)(x(i) - y(i)) - \frac{(x(i) - y(i))^2}{2x(i)}$$

$$\leq \frac{1}{2} \sum_{i=1}^{n} l(i)^2 x(i)$$

∎

Compare the result of Lemma 19 and Lemma 21. In Lemma 21, the inequality $l^{\top}(x - y) - \mathrm{B}_F(y\|x) \leq \frac{1}{2} \sum_{i=1}^{n} x(i)l(i)^2$ holds for all $l \in \mathbb{R}^n$. In Lemma 19, we have a slighly tighter bound $l^{\top}(x - y) - \mathrm{B}_F(y\|x) \leq \sum_{i=1}^{n} x(i)^2 l(i)^2$, which holds only if $\sum_{i=1}^{n} x(i)^2 l(i)^2 \leq \frac{1}{4}$. These results are summarized in Table 6.

Table 6: Upper bounds for $l^{\top}(x - y) - \mathrm{B}_F(y\|x)$ when $F(x) = -n\log(n) - \sum_{i=1}^{n} \log(x(i))$

| **Lemma** | **Domain of** $x, y$ | **Condition on** $l$ | **Upper bound** |
|---|---|---|---|
| Lemma 19 | $\mathbb{R}_+^n$ | $\|l\|_{\nabla^2 F(x)^{-1}} \leq 1/2$ | $\sum_{i=1}^{n} x(i)^2 l(i)^2$ |
| Lemma 21 | $\Delta_n$ | $l \in \mathbb{R}^n$ | $\frac{1}{2} \sum_{i=1}^{n} x(i)l(i)^2$ |

## A.4. Conclusion

We provided a general regret inequality for Optimistic FTRL in Theorem 12 from which we can obtain the regret bounds of all the algorithms in this paper. For tuning the learning rates in our algorithms, we use the general technique of AdaFTRL and obtain a bound on the regret using Lemma 15. Finally, as we use the log-barrier regularizer in some of our algorithms, we provide two techniques for bounding the stability term $\Psi_x(l)$ under different conditions. The first uses the theory of self-concordant function and is presented in Lemma 19. The second a different technique of obtaining local-norm lower bound and is presented in Lemma 21.

## Appendix B.   Proofs from Appendix A

We begin by stating a few useful properties of Bregman Divergences.

**Lemma 22**   *For any $v, w \in dom(\nabla F)$ and $u \in dom(F)$ we have:*

$$B_F(u\|w) - B_F(u\|v) - B_F(v\|w) = (\nabla F(w) - \nabla F(v))^\top (v - u)$$

Lemma 22 is called the *Law of cosines for Bregman divergence*.  The proof is via a direct calculation. The law of cosines can be extended to the case of Mixed Bregmans as well.

**Lemma 23**   *For any $v, w \in dom(\nabla F)$ and $u \in dom(F)$ we have:*

$$B_F^{a,b}(u\|w) - B_F^{a,c}(u\|v) - B_F^{c,b}(v\|w) = \left(\frac{\nabla F(w)}{b} - \frac{\nabla F(v)}{c}\right)^\top (v - u)$$

We now prove the main theorem obtaining a general regret bound for Optimistic FTRL.

**Theorem 12**   *For any $w \in \mathcal{D}$, any sequence of convex cost functions $f_1, \ldots, f_T$, convex hint functions $m_1, \ldots, m_T$, convex regularizer $F$ and parameters $\eta_0, \ldots, \eta_T$ such that $w_t \in \arg\min_{w \in \mathcal{D}} \sum_{s=1}^{t-1} f_s(w) + m_t(w) + \frac{F(w)}{\eta_{t-1}}$ and $w_t' \in \arg\min_{w \in \mathcal{D}} \sum_{s=1}^{t-1} f_s(w) + \frac{F(w)}{\eta_{t-1}}$. Let $g_t = \sum_{s=1}^{t} f_s$. The iterates of Optimistic FRTL $w_1, \ldots, w_T$ satisfies the regret inequality $\sum_{t=1}^{T} f_t(w_t) - f_t(w)$:*

$$\leq B_F^{\eta_T, \eta_0}(w\|w_1') + \sum_{t=1}^{T} \Big[(\nabla f_t(w_t) - \nabla m_t(w_t))^\top (w_t - w_{t+1}') - B_{g_t}(w_{t+1}'\|w_t) - B_F^{\eta_t, \eta_{t-1}}(w_{t+1}'\|w_t)$$

$$- B_{g_{t-1}}(w_t\|w_t') - B_F^{\eta_{t-1}, \eta_{t-1}}(w_t\|w_t')\Big]$$

*Further, if $F$ is such that $\min_{w \in \mathcal{D}} F(w) = 0$ and the sequence $\eta_0, \ldots, \eta_T$ is non-increasing, then the above bound simplifies to $\sum_{t=1}^{T} f_t(w_t) - f_t(w)$:*

$$\leq \frac{F(w)}{\eta_T} + \sum_{t=1}^{T} \Big[(\nabla f_t(w_t) - \nabla m_t(w_t))^\top (w_t - w_{t+1}') - B_{g_t}(w_{t+1}'\|w_t) - \frac{B_F(w_{t+1}'\|w_t)}{\eta_{t-1}}$$

$$- B_{g_{t-1}}(w_t\|w_t') - \frac{B_F(w_t\|w_t')}{\eta_{t-1}}\Big]$$

**Proof** Consider $f_t(w_t) - f_t(w)$. We expand it by adding and subtracting $f_t(w_{t+1}')$.

$$f_t(w_t) - f_t(w) = f_t(w_t) - f_t(w_{t+1}') + f_t(w_{t+1}') - f_t(w)$$

Using the definition of Bregman Divergence $B_{f_t}$

$$= \nabla f_t(w_t)^\top (w_t - w_{t+1}') - B_{f_t}(w_{t+1}'\|w_t) + \underbrace{\nabla f_t(w_{t+1}')^\top (w_{t+1}' - w)}_{(1)} - B_{f_t}(w\|w_{t+1}')$$

Consider term (1) in the above equation. Let $g_t(w) = \sum_{s=1}^{t} f_s(w)$. We can write $f_t(w) = g_t(w) - g_{t-1}(w)$, so $\nabla f_t(w) = \nabla g_t(w) - \nabla f_{t-1}(w)$. Substituting this expression for $\nabla f_t(w_{t+1}')$

$$(1) = \nabla f_t(w_{t+1}')^\top (w_{t+1}' - w) = (\nabla g_t(w_{t+1}') - \nabla g_{t-1}(w_{t+1}'))^\top (w_{t+1}' - w)$$

Adding and subtracting $\nabla g_{t-1}(w_t)^\top (w'_{t+1} - w)$

$$= (\nabla g_t(w'_{t+1}) - \nabla g_{t-1}(w_t))^\top (w'_{t+1} - w) + \underbrace{(\nabla g_{t-1}(w_t) - \nabla g_{t-1}(w'_{t+1}))^\top (w'_{t+1} - w)}_{(2)}$$

Using Lemma 22, the term (2) is:

$$(\nabla g_{t-1}(w_t) - \nabla g_{t-1}(w'_{t+1}))^\top (w'_{t+1} - w) = \mathbf{B}_{g_{t-1}}(w\|w_t) - \mathbf{B}_{g_{t-1}}(w\|w'_{t+1}) - \mathbf{B}_{g_{t-1}}(w'_{t+1}\|w_t)$$

Substituting this back in the expression for $f_t(w_t) - f_t(w)$ and rearranging, we have:

$$
\begin{aligned}
f_t(w_t) - f_t(w) &= \nabla f_t(w_t)^\top (w_t - w'_{t+1}) - \mathbf{B}_{f_t}(w'_{t+1}\|w_t) \\
&\quad + (\nabla g_t(w'_{t+1}) - \nabla g_{t-1}(w_t))^\top (w'_{t+1} - w) \\
&\quad + \mathbf{B}_{g_{t-1}}(w\|w_t) - \mathbf{B}_{g_{t-1}}(w\|w'_{t+1}) - \mathbf{B}_{g_{t-1}}(w'_{t+1}\|w_t) \\
&\quad - \mathbf{B}_{f_t}(w\|w'_{t+1}) \\
&= \nabla f_t(w_t)^\top (w_t - w'_{t+1}) - \mathbf{B}_{g_t}(w'_{t+1}\|w_t) \\
&\quad + (\nabla g_t(w'_{t+1}) - \nabla g_{t-1}(w_t))^\top (w'_{t+1} - w) \\
&\quad + \mathbf{B}_{g_{t-1}}(w\|w_t) - \mathbf{B}_{g_t}(w\|w'_{t+1}) \\
&= (\nabla f_t(w_t) - \nabla m_t(w_t))^\top (w_t - w'_{t+1}) - \mathbf{B}_{g_t}(w'_{t+1}\|w_t) \\
&\quad + (\nabla g_t(w'_{t+1}) - \nabla g_{t-1}(w_t) - \nabla m_t(w_t))^\top (w'_{t+1} - w) \\
&\quad + \mathbf{B}_{g_{t-1}}(w\|w_t) - \mathbf{B}_{g_t}(w\|w'_{t+1}) - \nabla m_t(w_t)^\top (w - w_t)
\end{aligned}
$$

By Lemma 22, we can write:

$$\mathbf{B}_{g_{t-1}}(w\|w_t) = \mathbf{B}_{g_{t-1}}(w\|w'_t) - \mathbf{B}_{g_{t-1}}(w_t\|w'_t) + (\nabla g_{t-1}(w'_t) - \nabla g_{t-1}(w_t))^\top (w - w_t)$$

Substituting this back in the expression for $f_t(w_t) - f_t(w)$, we and simplifying, we have:

$$
\begin{aligned}
f_t(w_t) - f_t(w) &= (\nabla f_t(w_t) - \nabla m_t(w_t))^\top (w_t - w'_{t+1}) - \mathbf{B}_{g_t}(w'_{t+1}\|w_t) - \mathbf{B}_{g_{t-1}}(w_t\|w'_t) \\
&\quad + \underbrace{(\nabla g_t(w'_{t+1}) - \nabla g_{t-1}(w_t) - \nabla m_t(w_t))^\top (w'_{t+1} - w)}_{(3)} \\
&\quad + \underbrace{(\nabla g_{t-1}(w_t) + \nabla m_t(w_t) - \nabla g_{t-1}(w'_t))^\top (w_t - w)}_{(4)} \\
&\quad + \mathbf{B}_{g_{t-1}}(w\|w'_t) - \mathbf{B}_{g_t}(w\|w'_{t+1})
\end{aligned}
$$

We introduce the following notation to ease the algebraic manipulation:

$$G_{t-1}(w) = g_{t-1}(w) + m_t(w) + \frac{F(w)}{\eta_{t-1}}$$

$$G'_t(w) = g_t(w) + \frac{F(w)}{\eta_t}$$

We simplify term (3) and apply Lemma 23:

$$(3) = (\nabla G'_t(w'_{t+1}) - \nabla G_{t-1}(w_t))^\top (w'_{t+1} - w) + \left( \frac{\nabla F(w_t)}{\eta_{t-1}} - \frac{\nabla F(w'_{t+1})}{\eta_t} \right)^\top (w'_{t+1} - w)$$

$$= (\nabla G'_t(w'_{t+1}) - \nabla G_{t-1}(w_t))^\top (w'_{t+1} - w) + \mathbf{B}_F^{\alpha,\eta_{t-1}}(w\|w_t) - \mathbf{B}_F^{\alpha,\eta_t}(w\|w'_{t+1}) - \mathbf{B}_F^{\eta_t,\eta_{t-1}}(w'_{t+1}\|w_t)$$

Similarly, we simplify term (4) and apply Lemma 23:

$$(4) = (\nabla G_{t-1}(w_t) - \nabla G'_{t-1}(w'_t))^\top (w_t - w) + \left( \frac{\nabla F(w'_t)}{\eta_{t-1}} - \frac{\nabla F(w_t)}{\eta_{t-1}} \right)^\top (w_t - w)$$

$$= (\nabla G_{t-1}(w_t) - \nabla G'_{t-1}(w'_t))^\top (w_t - w) + \mathbf{B}_F^{\alpha,\eta_{t-1}}(w\|w'_t) - \mathbf{B}_F^{\alpha,\eta_{t-1}}(w\|w_t) - \mathbf{B}_F^{\eta_{t-1},\eta_{t-1}}(w_t\|w'_t)$$

Substituting these back in the expression for $f_t(w_t) - f_t(w)$, we have:

$$= (\nabla f_t(w_t) - \nabla m_t(w_t))^\top (w_t - w'_{t+1}) - \mathbf{B}_{g_t}(w'_{t+1}\|w_t) - \mathbf{B}_{g_{t-1}}(w_t\|w'_t)$$
$$+ (\nabla G'_t(w'_{t+1}) - \nabla G_{t-1}(w_t))^\top (w'_{t+1} - w) + \mathbf{B}_F^{\eta_{t-1}}(w\|w_t) - \mathbf{B}_F^{\alpha,\eta_t}(w\|w'_{t+1}) - \mathbf{B}_F^{\eta_t,\eta_{t-1}}(w'_{t+1}\|w_t)$$
$$+ (\nabla G_{t-1}(w_t) - \nabla G'_{t-1}(w'_t))^\top (w_t - w) + \mathbf{B}_F^{\alpha,\eta_{t-1}}(w\|w'_t) - \mathbf{B}_F^{\alpha,\eta_{t-1}}(w\|w_t) - \mathbf{B}_F^{\eta_{t-1},\eta_{t-1}}(w_t\|w'_t)$$
$$+ \mathbf{B}_{g_{t-1}}(w\|w'_t) - \mathbf{B}_{g_t}(w\|w'_{t+1})$$
$$= (\nabla f_t(w_t) - \nabla m_t(w_t))^\top (w_t - w'_{t+1}) - \mathbf{B}_{g_t}(w'_{t+1}\|w_t) - \mathbf{B}_{g_{t-1}}(w_t\|w'_t)$$
$$- \mathbf{B}_F^{\eta_t,\eta_{t-1}}(w'_{t+1}\|w_t) - \mathbf{B}_F^{\eta_{t-1},\eta_{t-1}}(w_t\|w'_t)$$
$$+ \mathbf{B}_{g_{t-1}}(w\|w'_t) - \mathbf{B}_{g_t}(w\|w'_{t+1}) + \mathbf{B}_F^{\alpha,\eta_{t-1}}(w\|w'_t) - \mathbf{B}_F^{\alpha,\eta_t}(w\|w'_{t+1})$$
$$+ (\nabla G'_{t-1}(w'_t) - \nabla G'_t(w'_{t+1}))^\top w$$
$$+ \nabla G_{t-1}(w_t)^\top (w_t - w'_{t+1})$$
$$+ \nabla G'_t(w'_{t+1})w'_{t+1} - \nabla G'_{t-1}(w'_t)^\top w_t$$

Since $w_t$ minimizes $G_{t-1}(w)$, we have $(\nabla G_{t-1}(w_t))^\top (w_t - w'_{t+1}) \leq 0$. Taking the summation over the $t$ terms $\sum_{t=1}^T f_t(w_t) - f_t(w)$, we have :

$$\leq \sum_{t=1}^T \left( (\nabla f_t(w_t) - \nabla m_t(w_t))^\top (w_t - w'_{t+1}) - \mathbf{B}_{g_t}(w'_{t+1}\|w_t) - \mathbf{B}_F^{\eta_t,\eta_{t-1}}(w'_{t+1}\|w_t) \right)$$

$$+ \sum_{t=1}^T \left( -\mathbf{B}_{g_{t-1}}(w_t\|w'_t) - \mathbf{B}_F^{\eta_{t-1},\eta_{t-1}}(w_t\|w'_t) \right)$$

$$+ \underbrace{\sum_{t=1}^T \mathbf{B}_{g_{t-1}}(w\|w'_t) - \mathbf{B}_{g_t}(w\|w'_{t+1})}_{(5)} + \underbrace{\sum_{t=1}^T \mathbf{B}_F^{\alpha,\eta_{t-1}}(w\|w'_t) - \mathbf{B}_F^{\alpha,\eta_t}(w\|w'_{t+1})}_{(6)}$$

$$+ \underbrace{\sum_{t=1}^T (\nabla G'_{t-1}(w'_t) - \nabla G'_t(w'_{t+1}))^\top w}_{(7)} + \underbrace{\sum_{t=1}^T \left( \nabla G'_t(w'_{t+1})w'_{t+1} - \nabla G'_{t-1}(w'_t)^\top w_t \right)}_{(8)}$$

We can telescope term (5) to get:

$$\sum_{t=1}^{T} \mathbf{B}_{g_{t-1}}(w\|w_t') - \mathbf{B}_{g_t}(w\|w_{t+1}') = \mathbf{B}_{g_0}(w\|w_1') - \mathbf{B}_{g_T}(w\|w_{T+1}') = 0 - \mathbf{B}_{g_T}(w\|w_{T+1}') \le 0$$

We can telescope term (6) to get:

$$\sum_{t=1}^{T} \mathbf{B}_F^{\alpha,\eta_{t-1}}(w\|w_t') - \mathbf{B}_F^{\alpha,\eta_t}(w\|w_{t+1}') = \mathbf{B}_F^{\alpha,\eta_0}(w\|w_1') - \mathbf{B}_F^{\alpha,\eta_T}(w\|w_{T+1}')$$

Taking $\alpha = \eta_T$, we have:

$$\mathbf{B}_F^{\eta_T,\eta_0}(w\|w_1') - \mathbf{B}_F^{\eta_T,\eta_T}(w\|w_{T+1}') \le \mathbf{B}_F^{\eta_T,\eta_0}(w\|w_1')$$

Term (7) can be telescoped as:

$$\sum_{t=1}^{T} (\nabla G_{t-1}'(w_t') - \nabla G_t'(w_{t+1}'))^\top w = (\nabla G_0'(w_1') - \nabla G_T'(w_{T+1}'))^\top w = -\nabla G_T(w_{T+1}')^\top w$$

The hint for round $T+1$ can be taken as $m_{T+1}(w) = 0$. We have $w_{T+1} = w_{T+1}'$. Finally for term (8):

$$\sum_{t=1}^{T} \nabla G_t'(w_{t+1}')^\top w_{t+1}' - \nabla G_{t-1}'(w_t')^\top w_t = \sum_{t=1}^{T-1} \nabla G_t'(w_{t+1}')^\top (w_{t+1}' - w_{t+1}) + \nabla G_T'(w_{T+1}')^\top w_{T+1}$$

$$\le \nabla G_T'(w_{T+1}')^\top w_{T+1}'$$

Here, we used the fact that $w_{t+1}'$ minimizes $G_t'(w)$. So $\nabla G_t'(w_{t+1}')^\top (w_{t+1}' - w) \le 0$ for all $w \in \mathcal{D}$. Combining the upper bounds for terms (7) and (8):

$$(7) + (8) \le \nabla G_T(w_{T+1}')^\top (w_{T+1}' - w) \le 0$$

Thus, we have the result $\sum_{t=1}^{T} f_t(w_t) - f_t(w) \le$:

$$\le \mathbf{B}_F^{\eta_T,\eta_0}(w\|w_1') + \sum_{t=1}^{T} \Big[ (\nabla f_t(w_t) - \nabla m_t(w_t))^\top (w_t - w_{t+1}') - \mathbf{B}_{g_t}(w_{t+1}'\|w_t) - \mathbf{B}_F^{\eta_t,\eta_{t-1}}(w_{t+1}'\|w_t)$$

$$- \mathbf{B}_{g_{t-1}}(w_t\|w_t') - \mathbf{B}_F^{\eta_{t-1},\eta_{t-1}}(w_t\|w_t') \Big]$$

Further, if $F$ is such that $min_{w\in\mathcal{D}} F(w) = 0$, then $w_1' \in min_{w\in\mathcal{D}} F(w)$. So, $\mathbf{B}_F^{\eta_T,\eta_0}(w\|w_1') \le \frac{F(w)}{\eta_T}$. If the sequence $\eta_0, \dots, \eta_T$ is non-increasing, then

$$\mathbf{B}_F^{\eta_t,\eta_{t-1}}(w_{t+1}'\|w_t) \ge \frac{1}{\eta_{t-1}} \mathbf{B}_F(w_{t+1}'\|w_t)$$

the above bound simplifies to $\sum_{t=1}^{T} f_t(w_t) - f_t(w)$:

$$\le \frac{F(w)}{\eta_T} + \sum_{t=1}^{T} \Big[ (\nabla f_t(w_t) - \nabla m_t(w_t))^\top (w_t - w_{t+1}') - \mathbf{B}_{g_t}(w_{t+1}'\|w_t) - \frac{\mathbf{B}_F(w_{t+1}'\|w_t)}{\eta_{t-1}}$$

$$- \mathbf{B}_{g_{t-1}}(w_t\|w_t') - \frac{\mathbf{B}_F(w_t\|w_t')}{\eta_{t-1}} \Big]$$

$\blacksquare$

## Appendix C. Proofs from Section 3 and Section 4

**Lemma 1** *For all $x, y \in \Delta_n, r_t \in \mathbb{R}^n_+$ such that $r_t^\top x, r_t^\top y > 0$, we have the inequality:*

$$-\log(r_t^\top x) \geq -\log(r_t^\top y) - \frac{r_t^\top(x-y)}{r_t^\top y} + \frac{r_t^\top y}{2\max_i r_t(i)}\left(\frac{r_t^\top(x-y)}{r_t^\top y}\right)^2$$

**Proof** When $r_t^\top x, r_t^\top y > 0$, we have $0 < \frac{r_t^\top x}{\max_i r_t(i)}, \frac{r_t^\top y}{\max_i r_t(i)} \leq 1$. We apply Corollary 20 (with $x = \frac{r_t^\top y}{\max_i r_t(i)}$ and $y = \frac{r_t^\top x}{\max_i r_t(i)}$) to obtain:

$$\frac{r_t^\top x}{r_t^\top y} - 1 - \log\left(\frac{r_t^\top x}{r_t^\top y}\right) \geq \frac{1}{2}\frac{(r_t^\top x - r_t^\top y)^2}{(\max_i r_t(i))(r_t^\top y)}$$

$$\implies -\log(r_t^\top x) \geq -\log(r_t^\top y) - \frac{r_t^\top(x-y)}{r_t^\top y} + \frac{r_t^\top y}{2\max_i r_t(i)}\left(\frac{r_t^\top(x-y)}{r_t^\top y}\right)^2$$

■

Let $I$ be the $n \times n$ identity matrix. We state the following lemma, which is a tighter version of Lemma 11 in Hazan et al. (2007)

**Lemma 24** *(Hazan et al., 2007, Lemma 11) Let $x_1, \ldots, x_t$ be a sequence of vectors in $\mathbb{R}^n$. Define $H_t = \epsilon I + \sum_{s=1}^t x_s x_s^\top$. Then, the following holds:*

$$\sum_{t=1}^T x_t^\top H_t^{-1} x_t \leq n\log\left(1 + \frac{\sum_{t=1}^T \|x_t\|_2^2}{n\epsilon}\right)$$

**Lemma 25** *For any $w_1, \ldots, w_T \in \Delta_n$, and $r_1, \ldots, r_T \in \mathbb{R}^n_+$ we have the inequality:*

$$\frac{1}{2}\sum_{t=1}^T \frac{r_t}{r_t^\top w_t}^\top\left(\sum_{s=1}^t \frac{r_s r_s^\top}{(r_s^\top w_s)(\max_i r_s(i))} + \epsilon I + \lambda I\right)^{-1}\frac{r_t}{r_t^\top w_t} \leq \frac{nR}{2}\log\left(1 + \frac{\sum_{t=1}^T \|\hat{r}_t\|_2^2}{n(\epsilon + \lambda)}\right)$$

*Here $R = \max_{t,i,j}\frac{r_t(i)}{r_t(j)}$ and $\hat{r}_s = \frac{r_s}{\sqrt{(r_s^\top w_s)(\max_i r_s(i))}}$*

**Proof** Let $\tilde{r}_t = \frac{r_t}{\max_i r_t(i)}$. We re-write the above expression with $\tilde{r}_t$ as:

$$\frac{1}{2}\sum_{t=1}^T \frac{1}{(\tilde{r}_t^\top w_t)^2}\tilde{r}_t^\top\left(\sum_{s=1}^t \frac{\tilde{r}_s\tilde{r}_s^\top}{\tilde{r}_s^\top w_s} + \epsilon I + \lambda I\right)^{-1}\tilde{r}_t$$

Now let $\hat{r}_t = \frac{\tilde{r}_t}{\sqrt{\tilde{r}_t^\top w_t}} = \frac{r_t}{\sqrt{(r_t^\top w_t)(\max_i r_t(i))}} = \frac{r_t}{r_t^\top w_t}\sqrt{\frac{r_t^\top w_t}{\max_i r_t(i)}}$

$$\frac{1}{2\min_t(\tilde{r}_t^\top w_t)}\sum_{t=1}^T \hat{r}_t\left(\sum_{s=1}^t \hat{r}_s\hat{r}_s^\top + \epsilon I + \lambda I\right)^{-1}\hat{r}_t$$

We have $\frac{1}{\min_t(\tilde{r}_t^\top w_t)} = \max_t \frac{1}{(\tilde{r}_t^\top w_t)} = \max_t \frac{\max_i r_t(i)}{r_t^\top w_t} \leq \max_t \frac{\max_i r_t(i)}{\min_i r_t(i)} = \max_{t,i,j} \frac{r_t(i)}{r_t(j)} = R.$
Using the so called Elliptical potential lemma (Lemma 24), we have the bound:

$$\sum_{t=1}^{T} \hat{r}_t \left( \sum_{s=1}^{t} \hat{r}_s \hat{r}_s^\top + \epsilon I + \lambda I \right)^{-1} \hat{r}_t \leq n \log \left( 1 + \frac{\sum_{t=1}^{T} \|\hat{r}_t\|_2^2}{n(\epsilon + \lambda)} \right)$$

This gives us the stated result. ∎

**Theorem 2** *For $w \in \Delta$, any sequence of returns $r_1, \ldots, r_T \in \mathbb{R}_+^n$, define $f_t(w) = -\log(r_t^\top w)$. With $\epsilon = 1$, AdaCurv ONS (Equation (2)) has the data-dependent regret bound:*

$$\sum_{t=1}^{T} f_t(w_t) - f_t(w) \leq \frac{1}{2} + \frac{nR}{2} \log (1 + TR)$$

*If we set $\epsilon = 0$, we get the data-dependent regret bound for AdaCurv FTAL:*

$$\sum_{t=1}^{T} f_t(w_t) - f_t(w) \leq R + \frac{nR}{2} \log (1 + T^2)$$

**Proof**
Recall the adaptive curvature surrogate function in Equation (1):

$$\tilde{f}_t(w) = f_t(w_t) + \nabla f_t(w_t)^\top (w - w_t) + \frac{r_t^\top w_t}{2(\max_i r_t(i))} (\nabla f_t(w_t)^\top (w - w_t))^2$$

Due to Lemma 1, we know that that $\tilde{f}_t(w) \leq f_t(w)$ for all $w \in \mathcal{D}$ and $\tilde{f}_t(w_t) = f_t(w_t)$. Thus,

$$\sum_{t=1}^{T} f_t(w_t) - f_t(w) \leq \sum_{t=1}^{T} \tilde{f}_t(w_t) - \tilde{f}_t(w)$$

Applying Corollary 13 and with the constant regularizer $\frac{\epsilon}{2}\left(\|w\|_2^2 - \frac{1}{n}\right)$, we get:

$$\sum_{t=1}^{T} \tilde{f}_t(w_t) - \tilde{f}_t(w) \leq \frac{\epsilon}{2}\|w\|_2^2 + \sum_{t=1}^{T} \nabla \tilde{f}_t(w_t)^\top (w_t - w_{t+1}) - \mathrm{B}_{\tilde{g}_t}(w_{t+1}\|w_t) - \frac{\epsilon}{2}\|w_{t+1} - w_t\|_2^2$$

Here $\tilde{g}_t = \sum_{s=1}^{t} \tilde{f}_s$. Note that $\nabla \tilde{f}_t(w_t) = \nabla f_t(w_t) = -\frac{r_t}{r_t^\top w_t}$. Since $\tilde{g}_t(w)$ is quadratic in $w$, we have $\mathrm{B}_{\tilde{g}_t}(w_{t+1}\|w_t)$

$$= \frac{1}{2}(w_{t+1}-w_t)^\top \nabla^2 \tilde{g}_t(w_t)(w_{t+1}-w_t) = \frac{1}{2}(w_{t+1}-w_t)^\top \left( \sum_{s=1}^{t} \frac{r_s r_s^\top}{(r_s^\top w_s)(\max_i r_s(i))} \right)(w_{t+1}-w_t)$$

Thus $\sum_{t=1}^{T} f_t(w_t) - f_t(w) \leq$

$$\frac{\epsilon}{2}\|w\|_2^2 + \sum_{t=1}^{T} -\frac{r_t}{r_t^\top w_t}^\top (w_t - w_{t+1}) - \frac{1}{2}(w_{t+1} - w_t)^\top \left( \sum_{s=1}^{t} \frac{r_s r_s^\top}{(r_s^\top w_s)(\max_i r_s(i))} + \epsilon I \right)(w_{t+1} - w_t)$$

Add and subtract $\frac{\lambda}{2}\|w_{t+1} - w_t\|_2^2$.

$$
= \frac{\epsilon}{2}\|w\|_2^2 + \sum_{t=1}^{T} -\frac{r_t}{r_t^\top w_t}^\top (w_t - w_{t+1}) - \frac{1}{2}(w_{t+1} - w_t)^\top \left( \sum_{s=1}^{t} \frac{r_s r_s^\top}{(r_s^\top w_s)(\max_i r_s(i))} + \epsilon I + \lambda I \right) (w_{t+1} - w_t)
$$
$$
+ \sum_{t=1}^{T} \frac{\lambda}{2}\|w_{t+1} - w_t\|_2^2
$$

Upper bound the quadratic function in $w_{t+1} - w_t$.

$$
\leq \frac{\epsilon}{2}\|w\|_2^2 + \frac{1}{2}\sum_{t=1}^{T} \frac{r_t}{r_t^\top w_t}^\top \left( \sum_{s=1}^{t} \frac{r_s r_s^\top}{(r_s^\top w_s)(\max_i r_s(i))} + \epsilon I + \lambda I \right)^{-1} \frac{r_t}{r_t^\top w_t} + \sum_{t=1}^{T} \frac{\lambda}{2}\|w_{t+1} - w_t\|_2^2
$$

Since $w, w_t \in \Delta_n$, we can bound $\|w_{t+1} - w_t\|_2^2 \leq 2$ and $\|w\|_2^2 \leq 1$. Using Lemma 25, we have:

$$
\sum_{t=1}^{T} f_t(w_t) - f_t(w) \leq \frac{\epsilon}{2} + \frac{nR}{2}\log\left( 1 + \frac{\sum_{t=1}^{T}\|\hat{r}_t\|_2^2}{n(\epsilon + \lambda)} \right) + \lambda T
$$

Since $\lambda \geq 0$ can be chosen arbitrarily, we have the bound:

$$
\sum_{t=1}^{T} f_t(w_t) - f_t(w) \leq \frac{\epsilon}{2} + \inf_{\lambda \geq 0}\left( \lambda T + \frac{nR}{2}\log\left( 1 + \frac{\sum_{t=1}^{T}\|\hat{r}_t\|_2^2}{n(\epsilon + \lambda)} \right) \right)
$$

Choosing $\epsilon = 1$, $\lambda = 0$ and noting that $\|\hat{r}_t\|_2^2 \leq nR$ gives the regret bound for AdaCurv ONS.

$$
\sum_{t=1}^{T} f_t(w_t) - f_t(w) \leq \frac{1}{2} + \frac{nR}{2}\log\left( 1 + TR \right)
$$

Setting $\epsilon = 0$ and $\lambda = R/T$ gives us the regret bound for AdaCurv FTAL.

$$
\sum_{t=1}^{T} f_t(w_t) - f_t(w) \leq R + \frac{nR}{2}\log\left( 1 + T^2 \right)
$$

∎

**Theorem 3** *For $w \in \Delta$, any sequence of returns $r_1, \ldots, r_T \in \mathbb{R}_+^n$, define $f_t(w) = -\log(r_t^\top w)$. If we set $\epsilon = 1$, we get the bound for LB-AdaCurv ONS:*

$$
\sum_{t=1}^{T} f_t(w_t) - f_t(w) \leq \frac{5}{2} + 2n\log T + \min\left( nR\log\left( 1 + RT \right), 2 + 2\sqrt{2nT\log(T)} \right)
$$

*If we set $\epsilon = 0$, we get the bound for LB-AdaCurv FTAL:*

$$
\sum_{t=1}^{T} f_t(w_t) - f_t(w) \leq 2 + 2n\log(T) + \min\left( 2R + nR\log\left( 1 + T^2 \right), 2 + 2\sqrt{2nT\log(T)} \right)
$$

**Proof** First, we decompose the regret into two terms:

$$\sum_{t=1}^{T} f_t(w_t) - f_t(w) = \sum_{t=1}^{T} f_t(w_t) - f_t(w^\gamma) + \sum_{t=1}^{T} f_t(w^\gamma) - f_t(w)$$

Here $w^\gamma = (1-\gamma)w + \gamma/n$. For the second term, we have:

$$\sum_{t=1}^{T} f_t(w^\gamma) - f_t(w) = \sum_{t=1}^{T} \log\left(\frac{r_t^\top w}{(1-\gamma)r_t^\top w + \gamma r_t^\top \frac{1}{n}}\right) \leq T\log\left(\frac{1}{1-\gamma}\right) \leq 2\gamma T$$

Here, we used the fact that when $\gamma \leq 1/2$, we have $\log\left(\frac{1}{1-\gamma}\right) \leq 2\gamma$. For the first term, we use the surrogate function property:

$$\sum_{t=1}^{T} f_t(w_t) - f_t(w^\gamma) \leq \sum_{t=1}^{T} \tilde{f}_t(w_t) - \tilde{f}_t(w^\gamma)$$

So, we have:

$$\sum_{t=1}^{T} f_t(w_t) - f_t(w) \leq \sum_{t=1}^{T} \tilde{f}_t(w_t) - \tilde{f}_t(w^\gamma) + 2\gamma T$$

The iterates of LB-AdaCurv ONS are given by:

$$w_t \in \arg\min_{w \in \Delta_n} \sum_{s=1}^{t-1} \tilde{f}_s(w) + \frac{\epsilon}{2}\|w\|_2^2 + \frac{1}{\eta_{t-1}} \sum_{i=1}^{n} -\log(w(i))$$

The updates can be viewed as an FTRL with time varying learning rate $\eta_{t-1}$ for regularizer $F(w) = \sum_{i=1}^{n}[\log(1/n) - \log(w(i))]$ and constant regularizer $\frac{\epsilon}{2}\left(\|w\|_2^2 - \frac{1}{n}\right)$. Here $F(w)$ is the log-barrier regularizer. Using Corollary 14 we have the regret bound $\sum_{t=1}^{T} \tilde{f}_t(w_t) - \tilde{f}_t(w^\gamma)$

$$\leq \frac{\epsilon}{2}\|w^\gamma\|_2^2 + \frac{F(w^\gamma)}{\eta_T} + \sum_{t=1}^{T} \nabla\tilde{f}_t(w_t)^\top(w_t - w_{t+1}) - \mathrm{B}_{\tilde{g}_t}(w_{t+1}\|w_t) - \frac{\epsilon}{2}\|w_{t+1} - w_t\|_2^2 - \frac{1}{\eta_{t-1}}\mathrm{B}_F(w_{t+1}\|w_t)$$

Here $\tilde{g}_t = \sum_{s=1}^{t} \tilde{f}_s$. We compute $\nabla\tilde{f}_t(w_t) = \nabla f_t(w_t) = -\frac{r_t}{r_t^\top w_t}$. Moreover, $\tilde{g}_t(w)$ is quadratic in $w$, we have $\mathrm{B}_{\tilde{g}_t}(w_{t+1}\|w_t)$

$$= \frac{1}{2}(w_{t+1}-w_t)^\top \nabla^2\tilde{g}_t(w_t)(w_{t+1}-w_t) = \frac{1}{2}(w_{t+1}-w_t)^\top \left(\sum_{s=1}^{t} \frac{r_s r_s^\top}{(r_s^\top w_s)(\max_i r_s(i))}\right)(w_{t+1}-w_t)$$

We bound $F(w^\gamma)$ below:

$$F(w^\gamma) = n\log(1/n) - (n-1)\log(\gamma/n) - \log((1-\gamma)+\gamma/n)$$
$$\leq n\log(1/n) - n\log(\gamma/n) = n\log(1/\gamma)$$

So for the first term, we have:

$$\frac{\epsilon}{2}\|w^\gamma\|_2^2 + \frac{F(w^\gamma)}{\eta_T} \le \frac{\epsilon}{2} + \frac{n}{\eta_T}\log\left(\frac{1}{\gamma}\right)$$

Define $M_t(\eta_{t-1})$ as

$$M_t(\eta_{t-1}) = \sup_{w \in \Delta_n} \nabla f_t(w_t)^\top(w_t - w) - \mathbf{B}_{\tilde{g}_t}(w\|w_t) - \frac{\epsilon}{2}\|w - w_t\|_2^2 - \frac{1}{\eta_{t-1}}\mathbf{B}_F(w\|w_t)$$

Let $w_t^\star$ be the optimal value of $w$ in the optimization. We pick $\eta_t$ as:

$$\eta_t = \frac{\alpha}{\beta + \sum_{s=1}^t M_s(\eta_{s-1})}$$

We bound the regret in in two different ways. Substituting $\eta_t$ in the regret inequality, we have:

$$\sum_{t=1}^T \tilde{f}_t(w_t) - \tilde{f}_t(w^\gamma) \le \frac{\epsilon}{2} + \frac{n}{\eta_T}\log\left(\frac{1}{\gamma}\right) + \sum_{t=1}^T M_t(\eta_{t-1})$$

$$= \frac{\epsilon}{2} + \frac{n\log(1/\gamma)\beta}{\alpha} + \left(\frac{n\log(1/\gamma)}{\alpha} + 1\right)\left(\sum_{t=1}^T M_t(\eta_{t-1})\right)$$

Observe that $M_t(\eta_{t-1})$ can be written as:

$$M_t(\eta_{t-1}) = \nabla f_t(w_t)^\top(w_t - w_t^\star) - \mathbf{B}_{\tilde{g}_t}(w_t^\star\|w_t) - \frac{\epsilon}{2}\|w_t^\star - w_t\|_2^2 - \frac{1}{\eta_{t-1}}\mathbf{B}_F(w_t^\star\|w_t)$$

Ignoring the last Bregman term.

$$\le \nabla f_t(w_t)^\top(w_t - w_t^\star) - \mathbf{B}_{\tilde{g}_t}(w_t^\star\|w_t) - \frac{\epsilon}{2}\|w_t^\star - w_t\|_2^2$$

Adding and subtracting $\frac{\lambda}{2}\|w_t^\star - w_t\|_2^2$.

$$= \nabla f_t(w_t)^\top(w_t - w_t^\star) - \mathbf{B}_{\tilde{g}_t}(w_t^\star\|w_t) - \frac{\epsilon+\lambda}{2}\|w_t^\star - w_t\|_2^2 + \frac{\lambda}{2}\|w_t^\star - w_t\|_2^2$$

Taking the sum, we have $\sum_{t=1}^T M_t(\eta_{t-1})$

$$\le \sum_{t=1}^T\left(\nabla f_t(w_t)^\top(w_t - w_t^\star) - \mathbf{B}_{\tilde{g}_t}(w_t^\star\|w_t) - \frac{\epsilon+\lambda}{2}\|w_t^\star - w_t\|_2^2\right) + \sum_{t=1}^T \frac{\lambda}{2}\|w_t^\star - w_t\|_2^2$$

$$\le \sum_{t=1}^T -\frac{r_t}{r_t^\top w_t}^\top(w_t - w_t^\star) - \frac{1}{2}(w_t^\star - w_t)^\top\left(\sum_{s=1}^t \frac{r_s r_s^\top}{(r_s^\top w_s)(\max_i r_s(i))} + \epsilon I + \lambda I\right)(w_t^\star - w_t)$$

$$+ \sum_{t=1}^T \frac{\lambda}{2}\|w_t^\star - w_t\|_2^2$$

$$\le \frac{\lambda}{2}\sum_{t=1}^T\|w_t^\star - w_t\|_2^2 + \frac{1}{2}\sum_{t=1}^T \frac{r_t}{(r_t^\top w_t)}^\top\left(\sum_{s=1}^t \frac{r_s r_s^\top}{(r_s^\top w_s)(\max_i r_s(i))} + \epsilon I + \lambda I\right)^{-1}\frac{r_t}{(r_t^\top w_t)}$$

Using Lemma 25 and using the fact that $\lambda \geq 0$ can be chosen arbitrarily, we have:

$$\sum_{t=1}^{T} M_t(\eta_{t-1}) \leq \inf_{\lambda \geq 0} \left( \lambda T + \frac{nR}{2} \log \left( 1 + \frac{\sum_{t=1}^{T} \|\hat{r}_t\|_2^2}{n(\epsilon + \lambda)} \right) \right)$$

Thus, we have the following bound:

$$\frac{F(w^\gamma)}{\eta_T} + \sum_{t=1}^{T} M_t(\eta_{t-1}) \leq \frac{\epsilon}{2} + \frac{n \log(1/\gamma)\beta}{\alpha} + \left( \frac{n \log(1/\gamma)}{\alpha} + 1 \right) \left( \inf_{\lambda \geq 0} \left( \lambda T + \frac{nR}{2} \log \left( 1 + \frac{\sum_{t=1}^{T} \|\hat{r}_t\|_2^2}{n(\epsilon + \lambda)} \right) \right) \right)$$

Using $\alpha = n \log T$, $\beta = 2n \log T$ and $\gamma = 1/T$, the above bound yields:

$$\sum_{t=1}^{T} \tilde{f}_t(w_t) - \tilde{f}_t(w^\gamma) \leq \frac{\epsilon}{2} + 2n \log T + 2 \inf_{\lambda \geq 0} \left( \lambda T + \frac{\sum_{t=1}^{T} \|\hat{r}_t\|_2^2}{n(\epsilon + \lambda)} \right) \tag{6}$$

The second way to bound regret is below. Observe that:

$$\begin{aligned}
M_t(\eta_{t-1}) &= \nabla f_t(w_t)^\top (w_t - w_t^\star) - \mathbf{B}_{\tilde{g}_t}(w_t^\star \| w_t) - \frac{\epsilon}{2} \|w_t^\star - w_t\| - \frac{1}{\eta_{t-1}} \mathbf{B}_F(w_t^\star \| w_t) \\
&\leq \nabla f_t(w_t)^\top (w_t - w_t^\star) - \frac{1}{\eta_{t-1}} \mathbf{B}_F(w_t^\star \| w_t) \\
&= \frac{1}{\eta_{t-1}} \left[ \eta_{t-1} \nabla f_t(w_t)^\top (w_t - w_t^\star) - \mathbf{B}_F(w_t^\star \| w_t) \right. \\
&= \frac{1}{\eta_{t-1}} \left[ \eta_{t-1} (\nabla f_t(w_t) + c\mathbf{1})^\top (w_t - w_t^\star) - \mathbf{B}_F(w_t^\star \| w_t) \right]
\end{aligned}$$

Here $c$ can be any arbitrary constant.

Using Lemma 19, we have the following bound if $\|\eta_{t-1}(\nabla f_t(w_t) + c\mathbf{1})\|_{\nabla^2 F^{-1}(w_t)}^2 \leq \frac{1}{4}$.

$$M_t(\eta_{t-1}) \leq \frac{1}{\eta_{t-1}} \|\eta_{t-1}(\nabla f_t(w_t) + c\mathbf{1})\|_{\nabla^2 F^{-1}(w_t)}^2 = \eta_{t-1} \|(\nabla f_t(w_t) + c\mathbf{1}) \circ w_t\|_2^2$$

Since $c$, can be an arbitrary constant, we have that if $\eta_{t-1}^2 \inf_c \|(\nabla f_t(w_t) + c\mathbf{1}) \circ w_t\|_2^2 \leq 1/4$ then:

$$M_t(\eta_{t-1}) \leq \inf_c \eta_{t-1} \|(\nabla f_t(w_t) + c\mathbf{1}) \circ w_t\|_2^2$$

Choosing $c = 0$, we have that if $\eta_{t-1}^2 \inf_c \|(\nabla f_t(w_t) + c\mathbf{1}) \circ w_t\|_2^2 \leq \eta_{t-1}^2 \|\nabla f_t(w_t) \circ w_t\|_2^2 \leq 1/4$, then:

$$M_t(\eta_{t-1}) \leq \inf_c \eta_{t-1} \|(\nabla f_t(w_t) + c\mathbf{1}) \circ w_t\|_2^2 \leq \eta_{t-1} \|\nabla f_t(w_t) \circ w_t\|_2^2 \leq \eta_{t-1}$$

Observe that $\eta_{t-1}^2 \|\nabla f_t(w_t) \circ w_t\|_2^2 \leq \frac{1}{4}$ holds if $\eta_{t-1} \leq \frac{1}{2}$. Thus, when $\eta_{t-1} \leq \frac{1}{2}$, we have $M_t(\eta_{t-1}) \leq \eta_{t-1} \leq \frac{1}{2}$ and $M_t(\eta_{t-1})/\eta_{t-1} \leq \inf_c \|(\nabla f_t(w_t) + c\mathbf{1}) \circ w_t\|_2^2 \leq \|\nabla f_t(w_t) \circ w_t\|_2^2 \leq 1$.

Applying Lemma 15, we have the following bound::

$$\sum_{t=1}^{T} \tilde{f}_t(w_t) - \tilde{f}_t(w^\gamma) \leq \frac{\epsilon}{2} + \frac{n}{\eta_T} \log\left(\frac{1}{\gamma}\right) + \sum_{t=1}^{T} M_t(\eta_{t-1})$$

$$\leq \frac{\epsilon}{2} + n \log(1/\gamma) \left(\frac{\beta}{\alpha} + \frac{1}{2\alpha}\right)$$

$$+ \frac{1}{2} + \sqrt{2 \sum_{t=1}^{T} \inf_c \|(\nabla f_t(w_t) + c\mathbf{1}) \circ w_t\|_2^2 \left(\frac{n \log(1/\gamma)}{\sqrt{\alpha}} + \sqrt{\alpha}\right)}$$

Using $\alpha = n \log T$, $\beta = 2n \log T$ and $\gamma = 1/T$, the above bound yields $\sum_{t=1}^{T} \tilde{f}_t(w_t) - \tilde{f}_t(w^\gamma) \leq$

$$\frac{\epsilon}{2} + \left(2n \log(T) + \frac{1}{2}\right) + \frac{1}{2} + 2\sqrt{2n \left(\sum_{t=1}^{T} \inf_c \|(\nabla f_t(w_t) + c\mathbf{1}) \circ w_t\|_2^2\right) \log(T)} \qquad (7)$$

Both Equation (6) and Equation (7) hold simultaneously. Combining them, we have the bound:

$$\sum_{t=1}^{T} f_t(w_t) - f_t(w) \leq 2 + \frac{\epsilon}{2} + 2n \log T + 2 \min\left[\inf_{\lambda \geq 0}\left(\lambda T + \frac{nR}{2} \log\left(1 + \frac{\sum_{t=1}^{T} \|\hat{r}_t\|_2^2}{n(\epsilon + \lambda)}\right)\right),\right.$$

$$\left. 1 + \sqrt{2n \left(\sum_{t=1}^{T} \inf_c \|(\nabla f_t(w_t) + c\mathbf{1}) \circ w_t\|_2^2\right) \log(T)}\right]$$

Using $\epsilon = 1$ and $\lambda = 0$, The regret of LB-AdaCurv ONS will be:

$$\sum_{t=1}^{T} f_t(w_t) - f_t(w) \leq \frac{5}{2} + 2n \log T + \min\left(nR \log(1 + RT), 2 + 2\sqrt{2nT \log(T)}\right)$$

Using $\epsilon = 0$ and $\lambda = R/T$, the regret of LB-AdaCurv FTAL algorithm:

$$\sum_{t=1}^{T} f_t(w_t) - f_t(w) \leq 2 + 2n \log(T) + \min\left(2R + nR \log(1 + T^2), 2 + 2\sqrt{2nT \log(T)}\right)$$

$\blacksquare$

# Appendix D. Proofs from Section 5 and Section 6

**Theorem 4** *For $w \in \Delta$, any sequence of returns $r_1, \ldots, r_T \in \mathbb{R}_+^n$, return prediction distributions $D_1, \ldots, D_T$, concave and strictly increasing utility function $U$ with a strictly decreasing first derivative $U'$, define $f_t(w) = -\log(r_t^\top w)$. The updates of OEU-LB-FTRL (Algorithm 2) satisfy the regret bound:*

$$\sum_{t=1}^{T} f_t(w_t) - f_t(w) \leq 2 + C(1 + 2n \log T) + 2\sqrt{2n \left(\sum_{t=1}^{T} \left\|\mathbb{E}_{r \sim D_t}[U'(r^\top w_t) r \circ w_t] - \frac{r_t \circ w_t}{r_t^\top w_t}\right\|_2^2\right) \log T}$$

*Where $C = 1 + \sup_x xU'(x)$. This implies the worst-case regret bound:*

$$\sum_{t=1}^{T} f_t(w_t) - f_t(w) \le 2 + C\left(1 + 2n\log T\right) + 2C\sqrt{2nT\log T}$$

*Moreover, if $U$ and $D_t$ are such that $\mathbb{E}_{r\sim D_t}[U'(r^\top w_t)r \circ w_t] = \frac{r_t \circ w_t}{r_t^\top w_t}$, then we have the regret bound:*

$$\sum_{t=1}^{T} f_t(w_t) - f_t(w) \le 2 + 2Cn\log T$$

**Proof** First, we decompose the regret into two terms:

$$\sum_{t=1}^{T} f_t(w_t) - f_t(w) = \sum_{t=1}^{T} f_t(w_t) - f_t(w^\gamma) + \sum_{t=1}^{T} f_t(w^\gamma) - f_t(w)$$

Here $w^\gamma = (1-\gamma)w + \gamma/n$. For the second term, we have:

$$\sum_{t=1}^{T} f_t(w^\gamma) - f_t(w) = \sum_{t=1}^{T} \log\left(\frac{r_t^\top w}{(1-\gamma)r_t^\top w + \gamma r_t^\top \frac{1}{n}}\right) \le T\log\left(\frac{1}{1-\gamma}\right) \le 2\gamma T$$

Here, we used the fact that when $\gamma \le 1/2$, we have $\log\left(\frac{1}{1-\gamma}\right) \le 2\gamma$. For the first term, we use convexity:

$$\sum_{t=1}^{T} f_t(w_t) - f_t(w^\gamma) \le \sum_{t=1}^{T} \nabla f_t(w_t)^\top (w_t - w^\gamma)$$

So, we have:

$$\sum_{t=1}^{T} f_t(w_t) - f_t(w) \le \sum_{t=1}^{T} \nabla f_t(w_t)^\top (w_t - w^\gamma) + 2\gamma T$$

Since OEU-LB-FTRL is an instance of optimistic FTRL on the gradients $\nabla f_t(w_t)$, we can apply Theorem 12 with hint function $m_t(w) = -\mathbb{E}_{r\sim D_t}[U(r^\top w)]$. This gives the regret inequality $\nabla f_t(w_t)^\top (w_t - w^\gamma)$:

$$\le \frac{F(w^\gamma)}{\eta_T} + \sum_{t=1}^{T} \left[ (\nabla f_t(w_t) - \nabla m_t(w_t))^\top (w_t - w'_{t+1}) - \frac{\mathbf{B}_F(w'_{t+1}\|w_t)}{\eta_{t-1}} \right]$$

$$\le \frac{F(w^\gamma)}{\eta_T} + \sum_{t=1}^{T} M_t(\eta_{t-1})$$

Note that $\nabla m_t(w) = -\mathbb{E}_{r\sim D_t}[U'(r^\top w)r]$ and $w'_t$ are the iterates of LB-FTRL with no hints. We bound $F(w^\gamma)$ below:

$$F(w^\gamma) = n\log(1/n) - (n-1)\log(\gamma/n) - \log((1-\gamma) + \gamma/n)$$
$$\le n\log(1/n) - n\log(\gamma/n) = n\log(1/\gamma)$$

Consider $M_t(\eta_{t-1})$. Assume the supremum in it's optimization occurs at $w_t^\star$.

$$M_t(\eta_{t-1}) = \left(\nabla f_t(w_t) + \mathbb{E}_{r \sim D_t}[U'(r^\top w_t)r]\right)^\top (w_t - w_t^\star) - \frac{1}{\eta_{t-1}}\mathbf{B}_F(w_t^\star \| w_t)$$

$$= \frac{1}{\eta_{t-1}}\left(\eta_{t-1}\left(\nabla f_t(w_t) + \mathbb{E}_{r \sim D_t}[U'(r^\top w_t)r]\right)^\top (w_t - w_t^\star) - \mathbf{B}_F(w_t^\star \| w_t)\right)$$

Applying Lemma 19, if $\left\|\eta_{t-1}\left(\nabla f_t(w_t) + \mathbb{E}_{r \sim D_t}[U'(r^\top w_t)r]\right)\right\|_{\nabla^2 F(w_t)^{-1}} \le 1/2$, then:

$$M_t(\eta_{t-1}) \le \frac{1}{\eta_{t-1}}\left\|\eta_{t-1}\left(\nabla f_t(w_t) + \mathbb{E}_{r \sim D_t}[U'(r^\top w_t)r]\right)\right\|^2_{\nabla^2 F(w_t)^{-1}}$$

$$= \eta_{t-1}\left\|\mathbb{E}_{r \sim D_t}[U'(r^\top w_t)r \circ w_t] - \frac{r_t \circ w_t}{r_t^\top w_t}\right\|^2_2$$

$$= \eta_{t-1}\left\|\mathbb{E}_{r \sim D_t}\left[(r^\top w_t)U'(r^\top w_t)\frac{r \circ w_t}{r^\top w_t}\right] - \frac{r_t \circ w_t}{r_t^\top w_t}\right\|^2_2$$

$$\le \eta_{t-1}(1 + \sup_x xU'(x))^2$$

Since $\|\nabla f_t(w_t) + \mathbb{E}_{r \sim D_t}[U'(r^\top w_t)r]\|_{\nabla^2 F(w_t)^{-1}} \le 1 + \sup_x xU'(x)$, we can ensure that Lemma 19 is applicable by picking $\eta_{t-1} \le (2(1 + \sup_x xU'(x)))^{-1}$. Thus, we have

$$M_t(\eta_{t-1}) \le \eta_{t-1}(1 + \sup_x xU'(x))^2 \le \frac{1 + \sup_x xU'(x)}{2} = \frac{C}{2}$$

$$\frac{M_t(\eta_{t-1})}{\eta_{t-1}} \le \left\|\mathbb{E}_{r \sim D_t}[U'(r^\top w_t)r \circ w_t] - \frac{r_t \circ w_t}{r_t^\top w_t}\right\|^2_2 \le (1 + \sup_x xU'(x))^2 = C^2$$

Applying Lemma 15 and picking $\gamma = 1/T$, we have the bound:

$$\frac{F(w^\gamma)}{\eta_T} + \sum_{t=1}^T M_t(\eta_{t-1}) \le C\left(2n \log T + \frac{1}{2}\right) + \frac{C}{2}$$

$$+ 2\sqrt{2n\left(\sum_{t=1}^T \left\|\mathbb{E}_{r \sim D_t}[U'(r^\top w_t)r \circ w_t] - \frac{r_t \circ w_t}{r_t^\top w_t}\right\|^2_2\right)\log T}$$

Thus, we have the first bound:

$$\sum_{t=1}^T f_t(w_t) - f_t(w) \le 2 + C\left(1 + 2n \log T\right) + 2\sqrt{2n\left(\sum_{t=1}^T \left\|\mathbb{E}_{r \sim D_t}[U'(r^\top w_t)r \circ w_t] - \frac{r_t \circ w_t}{r_t^\top w_t}\right\|^2_2\right)\log T}$$

Since $\left\|\mathbb{E}_{r \sim D_t}[U'(r^\top w_t)r \circ w_t] - \frac{r_t \circ w_t}{r_t^\top w_t}\right\|^2_2 \le C^2$, this implies the worst-case bound:

$$\sum_{t=1}^T f_t(w_t) - f_t(w) \le 2 + C\left(1 + 2n \log T\right) + 2C\sqrt{2nT \log T}$$

If the condition $\mathbb{E}_{r \sim D_t}[U'(r^\top w_t) r \circ w_t] = \frac{r_t \circ w_t}{r_t^\top w_t}$ is satisfied, then $M_t(\eta_{t-1}) = 0$. In this case, we have the bound:

$$\sum_{t=1}^{T} f_t(w_t) - f_t(w) \leq 2 + \frac{F(w^\gamma)}{\eta_T} \leq 2 + 2Cn\log T$$

$\blacksquare$

**Theorem 5** *For $w \in \Delta_n$, any sequence of returns $r_1, \ldots, r_T \in \mathbb{R}_+^n$, return prediction distributions $D_1, \ldots, D_T$, concave and strictly increasing utility function $U$ with a strictly decreasing first derivative $U'$, define $f_t(w) = -\log(r_t^\top w)$. The updates of BoB-OPS (Algorithm 3) satisfy the regret bounds:*

$$\sum_{t=1}^{T} f_t(w_t) - f_t(w_t^{EU}) \leq \mathcal{R}_{metaRM}(2, T) \quad and \quad \sum_{t=1}^{T} f_t(w_t) - f_t(w) \leq \mathcal{R}_{RM}(n, T) + \mathcal{R}_{metaRM}(2, T)$$

*$\mathcal{R}_{RM}(n, T)$ and $\mathcal{R}_{metaRM}(2, T)$ are the regret bounds for the algorithm used by the RM investor and the meta-investor respectively. If the RM investor and meta-RM investor use Cover's Universal Portfolio (Cover, 1991) algorithm satisfy the regret bounds:*

$$\sum_{t=1}^{T} f_t(w_t) - f_t(w_t^{EU}) \leq \log(T+1) \quad and \quad \sum_{t=1}^{T} f_t(w_t) - f_t(w) \leq n\log(T+1)$$

**Proof** The regret bound for the meta-investor is:

$$\sum_{t=1}^{T} -\log(\gamma_t r_t^{EU} + (1-\gamma_t) r_t^{RM}) - \sum_{t=1}^{T} -\log(\gamma r_t^{EU} + (1-\gamma) r_t^{RM}) \leq \mathcal{R}_{metaRM}(2, T)$$

Pick $\gamma = 1$ and note that $r_t^{EU} = r_t^\top w_t^{EU}$, $r_t^{RM} = r_t^\top w_t^{RM}$

$$\implies \sum_{t=1}^{T} -\log(\gamma_t r_t^\top w_t^{EU} + (1-\gamma_t) r_t^\top w_t^{RM}) - \sum_{t=1}^{T} -\log(r_t^\top w_t^{EU}) \leq \mathcal{R}_{metaRM}(2, T)$$

$$\implies \sum_{t=1}^{T} -\log(r_t^\top w_t) - \sum_{t=1}^{T} -\log(r_t^\top w_t^{EU}) \leq \mathcal{R}_{metaRM}(2, T)$$

$$\implies \sum_{t=1}^{T} f_t(w_t) - \sum_{t=1}^{T} f_t(w_t^{EU}) \leq \mathcal{R}_{metaRM}(2, T)$$

This gives the first bound in the theorem. If we pick $\gamma = 0$, we would have arrived at:

$$\sum_{t=1}^{T} f_t(w_t) - \sum_{t=1}^{T} f_t(w_t^{RM}) \leq \mathcal{R}_{metaRM}(2, T)$$

The regret bound for the RM investor is:

$$\sum_{t=1}^{T} f_t(w_t^{RM}) - \sum_{t=1}^{T} f_t(w) \leq \mathcal{R}_{RM}(n, T)$$

Adding these two inequalities, we have the second bound in the theorem:

$$\sum_{t=1}^{T} f_t(w_t) - \sum_{t=1}^{T} f_t(w) \leq \mathcal{R}_{metaRM}(2, T) + \mathcal{R}_{RM}(n, T)$$

As the regret of Cover's UP algorithm is bounded by $(n-1)\log(T+1)$ (Cesa-Bianchi and Lugosi, 2006, Theorem 10.3), we get the final result stated in the theorem. ∎

## Appendix E. Proofs from Section 7

**Lemma 26** *(Srebro et al., 2010, Lemma 3.1) If a non-negative function $f$ is $H$-smooth on the domain $\mathcal{D}$, then $\|\nabla f(w)\| \leq \sqrt{4Hf(w)}$ for all $w \in \mathcal{D}$*

**Lemma 27** *Let $f_t = -\log(r_t^\top w)$ and let $w_t^\star = \arg\min_{w \in \Delta_n} f_t(w)$, i.e., it is the optimal portfolio for the return vector $r_t$. Let $l_t(w) = f_t(w) - f_t(w_t^\star)$. Then $l_t$ is $nR^2$-smooth on $\Delta_n$. So,*

$$\|\nabla f_t(w)\|_2^2 \leq 4nR^2 l_t(w)$$

**Proof** We have $\nabla l_t(w) = \nabla f_t(w)$. So for any $w, w' \in \Delta_n$, we have:

$$s\|\nabla l_t(w) - \nabla l_t(w')\|_2 = \left\| \frac{r_t}{r_t^\top w} - \frac{r_t}{r_t^\top w'} \right\|_2 = \frac{\|r_t\|_2^2 \|w - w'\|_2}{(r_t^\top w)(r_t^\top w')} \leq nR^2 \|w - w'\|_2$$

Thus $l_t(w)$ is $nR^2$ smooth on $\Delta_n$. Applying Lemma 26, we have the final result. ∎

**Lemma 28** *(Tsai et al., 2023b, Lemma 4.7) Let $f_t = -\log(r_t^\top w)$ and let $w_t^\star = \arg\min_{w \in \Delta_n} f_t(w)$, i.e., it is the optimal portfolio for the return vector $r_t$. Let $l_t(w) = f_t(w) - f_t(w_t^\star)$. We have,*

$$\inf_c \|(\nabla f_t(w) + c\boldsymbol{I}) \circ w\|_2^2 \leq 4l_t(w)$$

**Lemma 29** *(Orabona et al., 2012, Corollary 5) Let $a, b, c, d, x > 0$ satisfy $x \leq a\log(bx + c) + d$, then:*

$$x \leq a\log\left(2\left(ab\log\left(\frac{2ab}{e}\right) + db + c\right)\right) + d$$

*Here $e$ is the base of the natural logarithm.*

**Lemma 30** *(Orabona, 2019, Lemma 4.24) Let $a, b, c, , x > 0$ satisfy $x \leq c + \sqrt{ax + b}$, then:*

$$x \leq a + c + 2\sqrt{b + ac}$$

**Theorem 6** *For $w \in \Delta$, any sequence of returns $r_1, \ldots, r_T \in \mathbb{R}_+^n$, define $f_t(w) = -\log(r_t^\top w)$. The updates of AdaCurv ONS (Equation (2)) with $\epsilon = 1$ satisfy the regret bound:*

$$\sum_{t=1}^{T} f_t(w_t) - f_t(w) \leq \frac{1}{2} + \frac{nR}{2}\log\left(4nR^3\log\left(\frac{4nR^3}{e}\right) + 4R^2 + 8R^2 L_T^\star + 2\right)$$

*Here, $L_T^\star = \min_{w \in \Delta_n}\left[\sum_{t=1}^{T} f_t(w)\right] - \sum_{t=1}^{T}\left[\min_{w \in \Delta_n} f_t(w)\right]$ is the regret between the best static and the best dynamic portfolio selection strategies.*

**Proof** Consider the result from Theorem 2:

$$\sum_{t=1}^{T} f_t(w_t) - f_t(w) \le \frac{\epsilon}{2} + \inf_{\lambda \ge 0} \left( \lambda T + \frac{nR}{2} \log \left( 1 + \frac{\sum_{t=1}^{T} \|\hat{r}_t\|_2^2}{n(\epsilon + \lambda)} \right) \right)$$

Note that :

$$\|\hat{r}_t\|_2^2 = \left\| \frac{r_t}{r_t^\top w_t} \sqrt{\frac{r_t^\top w_t}{\max_i r_t(i)}} \right\|_2^2 = \frac{r_t^\top w_t}{\max_i r_t(i)} \|\nabla f_t(w_t)\|_2^2 \le \|\nabla f_t(w_t)\|_2^2$$

Let $w_t^\star = \arg\min_{w \in \Delta_n} f_t(w)$, i.e., it is the optimal portfolio for the return vector $r_t$. Let $l_t(w) = f_t(w) - f_t(w_t^\star)$. Note that $\nabla f_t(w) = \nabla l_t(w)$. So, we have:

$$\sum_{t=1}^{T} f_t(w_t) - f_t(w) \le \frac{\epsilon}{2} + \inf_{\lambda \ge 0} \left( \lambda T + \frac{nR}{2} \log \left( 1 + \frac{\sum_{t=1}^{T} \|\nabla f_t(w_t)\|_2^2}{n(\epsilon + \lambda)} \right) \right)$$

Pick $\epsilon = 1$ and $\lambda = 0$:

$$\sum_{t=1}^{T} f_t(w_t) - f_t(w) \le \frac{1}{2} + \frac{nR}{2} \log \left( 1 + \frac{\sum_{t=1}^{T} \|\nabla f_t(w_t)\|_2^2}{n} \right)$$

Apply Lemma 27, which states that $\|\nabla f_t(w_t)\|_2^2 \le 4nR^2 l_t(w_t) = 4nR^2 \left( f_t(w_t) - f_t(w_t^\star) \right)$:

$$\sum_{t=1}^{T} f_t(w_t) - f_t(w) \le \frac{1}{2} + \frac{nR}{2} \log \left( 1 + 4R^2 \left( \sum_{t=1}^{T} f_t(w_t) - f_t(w_t^\star) \right) \right)$$

$$\sum_{t=1}^{T} f_t(w_t) - f_t(w_t^\star) \le \sum_{t=1}^{T} f_t(w) - f_t(w_t^\star) + \frac{1}{2} + \frac{nR}{2} \log \left( 1 + 4R^2 \left( \sum_{t=1}^{T} f_t(w_t) - f_t(w_t^\star) \right) \right)$$

Now, we apply Lemma 29 with $a = nR/2$, $b = 4R^2$, $c = 1$, $d = 1/2 + \sum_{t=1}^{T} f_t(w) - f_t(w_t^\star)$ and $x = \sum_{t=1}^{T} f_t(w_t) - f_t(w_t^\star)$. So, we have the inequality:

$$\sum_{t=1}^{T} f_t(w_t) - f_t(w_t^\star) \le \frac{1}{2} + \sum_{t=1}^{T} f_t(w) - f_t(w_t^\star)$$
$$+ \frac{nR}{2} \log \left( 4nR^3 \log \left( \frac{4nR^3}{e} \right) + 4R^2 + 8R^2 \left( \sum_{t=1}^{T} f_t(w) - f_t(w_t^\star) \right) + 2 \right)$$

$$\sum_{t=1}^{T} f_t(w_t) - f_t(w) \le \frac{1}{2} + \frac{nR}{2} \log \left( 4nR^3 \log \left( \frac{4nR^3}{e} \right) + 4R^2 + 8R^2 \left( \sum_{t=1}^{T} f_t(w) - f_t(w_t^\star) \right) + 2 \right)$$

Specifically, if we pick $w^\star \in \arg\min_{w \in \Delta_n} \sum_{t=1}^{T} f_t(w)$. Then, $\sum_{t=1}^{T} f_t(w^\star) - f_t(w_t^\star)$ is the regret between the best static and the best dynamic portfolio selection strategies. We use the shorthand $L_T^\star$ to denote this quantity. Thus, we have the bound:

$$\sum_{t=1}^{T} f_t(w_t) - f_t(w) \le \frac{1}{2} + \frac{nR}{2} \log \left( 4nR^3 \log \left( \frac{4nR^3}{e} \right) + 4R^2 + 8R^2 L_T^\star + 2 \right)$$

∎

**Theorem 7** *For $w \in \Delta$, any sequence of returns $r_1, \ldots, r_T \in \mathbb{R}^n_+$, define $f_t(w) = -\log(r_t^\top w)$. The updates of LB-AdaCurv ONS (Algorithm 1) with $\epsilon = 1$ satisfy the regret bound:*

$$\sum_{t=1}^{T} f_t(w_t) - f_t(w_t) \leq \frac{5}{2} + 2n \log T + \min \left[ 2 + 2\sqrt{8n \log T} + 4\sqrt{8n \left( L_T^\star + \frac{9}{2} + 2n \log T \right) \log T}, \right.$$

$$\left. nR \log \left( 8nR^3 \log \left( \frac{8nR^3}{e} \right) + 20R^2 + 16R^2 n \log T + 8R^2 L_T^\star + 2 \right) \right]$$

*Here, $L_T^\star = \min_{w \in \Delta_n} \left[ \sum_{t=1}^{T} f_t(w) \right] - \sum_{t=1}^{T} \left[ \min_{w \in \Delta_n} f_t(w) \right]$ is the regret between the best static and the best dynamic portfolio selection strategies.*

**Proof** From Theorem 3, we have the following bound for LB-AdaCurv ONS after picking $\epsilon = 1$ and $\lambda = 0$:

$$\sum_{t=1}^{T} f_t(w_t) - f_t(w) \leq \frac{5}{2} + 2n \log T + 2 \min \left[ \frac{nR}{2} \log \left( 1 + \frac{\sum_{t=1}^{T} \|\hat{r}_t\|_2^2}{n} \right), \right.$$

$$\left. 1 + \sqrt{2n \left( \sum_{t=1}^{T} \inf_c \|(\nabla f_t(w_t) + c\mathbf{1}) \circ w_t\|_2^2 \right) \log(T)} \right]$$

Consider just the first part of the minimum. Note that :

$$\|\hat{r}_t\|_2^2 = \left\| \frac{r_t}{r_t^\top w_t} \sqrt{\frac{r_t^\top w_t}{\max_i r_t(i)}} \right\|_2^2 = \frac{r_t^\top w_t}{\max_i r_t(i)} \|\nabla f_t(w_t)\|_2^2 \leq \|\nabla f_t(w_t)\|_2^2$$

Let $w_t^\star = \arg\min_{w \in \Delta_n} f_t(w)$, i.e., it is the optimal portfolio for the return vector $r_t$. Let $l_t(w) = f_t(w) - f_t(w_t^\star)$. Note that $\nabla f_t(w) = \nabla l_t(w)$. So, we have:

$$\sum_{t=1}^{T} f_t(w_t) - f_t(w) \leq \frac{5}{2} + 2n \log T + nR \log \left( 1 + \frac{\sum_{t=1}^{T} \|\nabla f_t(w_t)\|_2^2}{n} \right)$$

Apply Lemma 27, which states that $\|\nabla f_t(w_t)\|_2^2 \leq 4nR^2 l_t(w_t) = 4nR^2 (f_t(w_t) - f_t(w_t^\star))$:

$$\sum_{t=1}^{T} f_t(w_t) - f_t(w) \leq \frac{5}{2} + 2n \log T + nR \log \left( 1 + 4R^2 \left( \sum_{t=1}^{T} f_t(w_t) - f_t(w_t^\star) \right) \right)$$

$$\sum_{t=1}^{T} f_t(w_t) - f_t(w_t^\star) \leq \sum_{t=1}^{T} f_t(w) - f_t(w_t^\star) + \frac{5}{2} + 2n \log T + nR \log \left( 1 + 4R^2 \left( \sum_{t=1}^{T} f_t(w_t) - f_t(w_t^\star) \right) \right)$$

Now, we apply Lemma 29 with $a = nR$, $b = 4R^2$, $c = 1$, $d = 5/2 + 2n \log T + \sum_{t=1}^{T} f_t(w) - f_t(w_t^\star)$ and $x = \sum_{t=1}^{T} f_t(w_t) - f_t(w_t^\star)$. So, we have the inequality:

$$\sum_{t=1}^{T} f_t(w_t) - f_t(w_t^\star) \leq \frac{5}{2} + 2n \log T + \sum_{t=1}^{T} f_t(w) - f_t(w_t^\star)$$

$$+ nR \log \left( 8nR^3 \log \left( \frac{8nR^3}{e} \right) + 20R^2 + 16R^2 n \log T + 8R^2 \left( \sum_{t=1}^{T} f_t(w) - f_t(w_t^\star) \right) + 2 \right)$$

$$\sum_{t=1}^{T} f_t(w_t) - f_t(w) \leq \frac{5}{2} + 2n \log T +$$

$$+ nR \log \left( 8nR^3 \log \left( \frac{8nR^3}{e} \right) + 20R^2 + 16R^2 n \log T + 8R^2 \left( \sum_{t=1}^{T} f_t(w) - f_t(w_t^\star) \right) + 2 \right)$$

Specifically, if we pick $w^\star \in \arg\min_{w \in \Delta_n} \sum_{t=1}^{T} f_t(w)$. Then, $L_T^\star = \sum_{t=1}^{T} f_t(w^\star) - f_t(w_t^\star)$ is the regret between the best static and the best dynamic portfolio selection strategies. We use the shorthand $L_T^\star$ to denote this quantity. Thus, we have the bound:

$$\sum_{t=1}^{T} f_t(w_t) - f_t(w) \leq \frac{5}{2} + 2n \log T + nR \log \left( 8nR^3 \log \left( \frac{8nR^3}{e} \right) + 20R^2 + 16R^2 n \log T + 8R^2 L_T^\star + 2 \right)$$

Consider the second part of the minimum:

$$\sum_{t=1}^{T} f_t(w_t) - f_t(w) \leq \frac{5}{2} + 2n \log T + 2 + 2\sqrt{2n \left( \sum_{t=1}^{T} \inf_c \|(\nabla f_t(w_t) + c\mathbf{1}) \circ w_t\|_2^2 \right) \log(T)}$$

Using Lemma 28, we have the bound $\inf_c \|(\nabla f_t(w_t) + c\mathbf{1}) \circ w_t\|_2^2 \leq 4(f_t(w_t) - f_t(w_t^\star))$:

$$\sum_{t=1}^{T} f_t(w_t) - f_t(w) \leq \frac{5}{2} + 2n \log T + 2 + 2\sqrt{8n \left( \sum_{t=1}^{T} f_t(w_t) - f_t(w_t^\star) \right) \log(T)}$$

$$\implies \sum_{t=1}^{T} f_t(w_t) - f_t(w_t^\star) \leq \sum_{t=1}^{T} f_t(w) - f_t(w_t^\star) + \frac{5}{2} + 2n \log T + 2 + 2\sqrt{8n \left( \sum_{t=1}^{T} f_t(w_t) - f_t(w_t^\star) \right) \log(T)}$$

Now, we apply Lemma 30 with $c = \sum_{t=1}^{T} f_t(w) - f_t(w_t^\star) + \frac{5}{2} + 2n \log T + 2$, $a = 2\sqrt{8n \log T}$, $b = 0$:

$$\sum_{t=1}^{T} f_t(w_t) - f_t(w_t^\star) \leq \sum_{t=1}^{T} f_t(w) - f_t(w_t^\star) + \frac{5}{2} + 2n \log T + 2 + 2\sqrt{8n \log T}$$

$$+ 4\sqrt{8n \log T \left( \sum_{t=1}^{T} f_t(w) - f_t(w_t^\star) + \frac{5}{2} + 2n \log T + 2 \right)}$$

$$\implies \sum_{t=1}^{T} f_t(w_t) - f_t(w) \leq \frac{9}{2} + 2n \log T + 2\sqrt{8n \log T} + 4\sqrt{8n \left( L_T^\star + \frac{9}{2} + 2n \log T \right) \log T}$$

Combining the two results, we get the final bound. ∎

In order to obtain the $O(\log Q_T)$ regret bound, we state a slightly modified version of a theorem from Hazan and Kale (2015).

**Theorem 31** *(Hazan and Kale, 2015, Theorem 1.1) Let the cost functions be $f_t(w) = h_t(w^\top v_t)$ for a scalar function $h_t$. Consider the iterates:*

$$w_t = \arg\min_{w \in \mathcal{D}} \frac{1}{2}\|w\|_2^2 + \sum_{s=1}^{t-1} h_s(w^\top v_s)$$

*If $\|v_t\| \le V$, $\|w\| \le D$ for all $w \in \mathcal{D}$, $h_t'(w_t^\top v_t) \in [-a, 0]$ and $h_t''(w^\top v_t) \ge b$ for all $w \in \mathcal{D}$, then:*

$$\mathcal{R}_T(w) \le O\left(\frac{a^2 n}{b}\log(1 + bQ_T + bV^2) + aVD\log(1 + Q_T/V^2) + D^2\right)$$

*Here $Q_T = \min_\mu \sum_{t=1}^{T} \|v_t - \mu\|$*

In the statement of the theorem in Hazan and Kale (2015), they assume that $h_t = h$ for all $t$ and $h'(w^\top v_t) \in [-a, 0]$ for all $w \in \mathcal{D}$. However, they later note that the proof of the theorem is flexible enough to handle different functions $h_t$ for different $t$. Furthermore, the proof only requires the bound $a$ on the magnitude of the first derivatives at the points $w_t$, which the algorithm produces, and not the entire domain $\mathcal{D}$.

**Theorem 8** *For $w \in \Delta$, any sequence of returns $r_1, \ldots, r_T \in \mathbb{R}_+^n$, define $f_t(w) = -\log(r_t^\top w)$. The AdaCurv ONS updates (Equation (2)) with $\epsilon = 1$ satisfy the regret bound:*

$$\sum_{t=1}^{T} f_t(w_t) - f_t(w) = O\left(nR^2\log(1 + Q_T + n) + \sqrt{n}R\log(1 + Q_T/n) + 1\right)$$

*Here $Q_T = \min_\mu \sum_{t=1}^{T} \|r_t - \mu\|_2^2 = \sum_{t=1}^{T} \|r_t - \bar{r}_T\|_2^2$, where $\bar{r}_T = \frac{1}{T}\sum_{t=1}^{T} r_t$.*

**Proof** The iterates of AdaCurvONS are computed as:

$$w_t = \arg\min_{w \in \Delta_n} \frac{1}{2}\|w\|_2^2 + \sum_{s=1}^{t-1}\left(f_s(w_s) - \frac{r_s^\top(w - w_s)}{(r_s^\top w_s)^\top} + \frac{(r_s^\top(w - w_s))^2}{2(r_s^\top w_s)}\right)$$

We can replace $r_t$ with $\tilde{r}_t = \frac{r_t}{\min_i r_t(i)}$ in the above equation without changing the iterates as the optimization is invariant to scaling. We can apply Theorem 31 with $v_t = \tilde{r}_t$. The function $h_t(x) = f_t(w_t) - \frac{x - \tilde{r}_s^\top w_s}{(\tilde{r}_s^\top w_s)^\top} + \frac{(x - \tilde{r}_s^\top w_s)^2}{2(\tilde{r}_s^\top w_s)}$. This gives $h_t'(\tilde{r}_t^\top w_t) = \frac{-1}{\tilde{r}_t^\top w_t} \in [-R, 0]$ and $h_t''(\tilde{r}_t^\top w) = \frac{1}{\tilde{r}_t^\top w_t} \ge 1$. Thus we have $\|\tilde{r}_t\| \le \sqrt{n} = V$, $D = 1$, $a = R$ and $b = 1$. So, we have the regret bound:

$$\sum_{t=1}^{T} f_t(w_t) - f_t(w) = O\left(nR^2\log(1 + Q_T + n) + \sqrt{n}R\log(1 + Q_T/n) + 1\right)$$

Here $Q_T = \min_\mu \sum_{t=1}^{T} \|r_t - \mu\| = \sum_{t=1}^{T} \|r_t - \bar{r}_T\|$, where $\bar{r}_T = \frac{1}{T}\sum_{t=1}^{T} r_t$. ∎

**Theorem 9** *For $w \in \Delta$, any sequence of returns $r_1, \ldots, r_T \in \mathbb{R}_+^n$, let the return prediction distribution $D_t$ be the delta distribution on $r_{t-1}$ (Let $r_0$ be the all 1s vector). The updates of OEU-LB-FTRL (Algorithm 2) with $U(x) = \log(x)$ satisfy the regret bound:*

$$\sum_{t=1}^{T} f_t(w_t) - f_t(w) \leq 4 + 4n \log T + 2\sqrt{2n\tilde{V}_T' \log T}$$

*Here $\tilde{V}_T' = \sum_{t=1}^{T} \left\| \frac{r_t \circ w_t}{r_t^\top w_t} - \frac{r_{t-1} \circ w_t}{r_{t-1}^\top w_t} \right\|_2^2$ and $r_0$ is the all ones vector.*

**Proof** Apply Theorem 4 with $U(x) = \log(x)$ and $D_t$ being the delta distribution on $r_{t-1}$. ∎

