# OpenReview forum: "Data Dependent Regret Bounds for Online Portfolio Selection with Predicted Returns"
_algorithmiclearningtheory.org/ALT/2025/Conference — ALT 2025_

### Official Review · Reviewer_GdQ4 · 2024-11-08
**Lots of new results for the classic OPS problem**

**Rating:** 7
**Confidence:** 4

**Review:**

The paper focuses on data-dependent regret bounds for the Online Portfolio Selection (OPS) problem with n assets and T time periods. They propose the LB-AdaCurv ONS algorithm with a worst-case regret of O(\sqrt{nT log T}) and a data-dependent regret of O(nR log T). The OUE-LB-FTRL algorithm achieves a regret of O(n log T) with accurate predictions and O(\sqrt{nT log T}) otherwise. The BoB-OPS meta-algorithm combines portfolios of an expected utility investor and a regret minimizing investor to achieve a regret of O(log T) for wrt the expected utility investor and O(n log T) for static regret. They also provide several new First-Order, Second-Order and Gradual-Variation regret bounds for OPS.

Overall, I found the topic of the paper to be relevant, the new problem variants to be interesting (eg: integrating predicted returns), the technical advances to be nontrivial, and thus argue for the paper to be accepted. Some slightly more detailed comments are below.

The first two sections were very well written and easy to read for a non-expert (though it is dense, and one must read every line!). From this, it is clear that the results (if correct) are quite extensive and impressive, and require algorithmic as well as analytic advances to accomplish. However, the paper is extremely long and I did not have the time to check the proofs. I appreciated the tables very much, these were great summaries.

The second half of the paper is well written enough to understand the algorithms, and the claimed theorems. But it is somewhat short of intuition. It is like a long list of algorithms and guarantees. In some sense this could be considered acceptable: the proofs are provided and one could check them, and the algorithms are variants of existing, sometimes well known, algorithms (with important tricks and alterations), so it would take too much space to describe the intuition of each algorithm. Nevertheless, it was not such an easy read.

I had a basic question. Why are some regret bounds of the form n \log T (which is minimax optimal) and others of the form \sqrt{T \log n}? What kinds of algorithmic techniques deliver regret logarithmic in n versus in T? Is there a fundamental reason that one cannot get "a best of both worlds" of this type?

**Paper Award:**

No

---

> ### Author Response · Authors · 2024-11-24
>
> Thank you for your review of our paper.
>
> Given the page limits, we decided on the current structure of the paper as the best way to present the results. A reader that is interested in knowing the results can just read the first two sections and understand the contributions. A reader interested in learning further about the algorithms and the tricks involved in obtaining the results can read the remaining sections.
>
> We appreciate the reviewer's effort in reading the terse later sections of the paper. As you correctly noted, our algorithms modify well known algorithms with important tricks. While a intuitive explanation of these tricks remains elusive to us, it is perhaps worth noting that intuition is subjective. Some readers might agree with our intuition while others might find it entirely confusing. So, we resorted to presenting our techniques in their rawest form as parts of theorems and lemmas, leaving no room for ambiguity.

---

> > ### Comment · Reviewer_GdQ4 · 2024-11-30
> > **Follow-up**
> >
> > Could you answer my question from the first review? Why are some regret bounds of the form n \log T (which is minimax optimal) and others of the form \sqrt{T \log n}? What kinds of algorithmic techniques deliver regret logarithmic in n versus in T? Is there a fundamental reason that one cannot get "a best of both worlds" of this type?
> >
> > (By best of both worlds I mean something like min(C_1 n \log T, C_2 \sqrt{T \log n}) where C_1 and C_2 are problem dependent parameters, and in the worst case these parameters would not allow the minimax lower bound to be violated but in the best case they may show some type of adaptive behavior between the two types of bounds)

---

> > ### Author Response · Authors · 2024-11-30
> >
> > Regret bounds of the form $\sqrt{T \log n}$ first linearize the loss function $- \log(r^\top w)$, then use an algorithm for "learning with expert advice" such as exponential weights or FTRL. Such algorithms typically get $\sqrt{T \log n}$ kind of regret bound.
> >
> > Techniques for obtaining $n \log T$ type regret bounds are more intricate. At the minimum, they need to use some form of second order information from the loss function.
> >
> > Getting a best-of-both worlds regret bound of the form you mentioned $\min(C_1 n \log T, C_2 \sqrt{T \log n}) $ is quite straight forward by using a meta algorithm similar to our BoB-OPS algorithm.
> >
> > There are 4 base investors. They run Cover's algorithm, Soft-Bayes, AdaHedge and AdaCurv ONS. The meta-investor who can only allocate to these 4 base investors runs Cover's algorithm.
> >
> > For this ensamble, the static-regret is bounded by $ 4\log T + \min(n \log T, \sqrt{nT \log n}, R \sqrt{T \log n}, R n \log T)$
> >
> > So, the regret for this ensamble is never vacuous because of the first two bases investors running Cover's algorithm and soft-bayes. It is also adaptive due to the last two base investors running AdaHedge and AdaCurv ONS.

---

### Official Review · Reviewer_k6ge · 2024-11-09
**Review of "Data Dependent Regret Bounds for Online Portfolio Selection with Predicted Returns"**

**Rating:** 7
**Confidence:** 3

**Review:**

This paper provides algorithms for online portfolio selection problem.
Proposed algorithms achieves data-dependent regret upper bound that is logarithmic in the number of time periods $T$,
as well as problem-independent regret upper bound of $\sqrt{T}$.
These results are expanded to derive first- and second-order regret bounds,
by incorporating optimistic online learning framework with convex hint functions.


One of the novel technical contributions in their paper is the construction of a new adaptive surrogate objective function to derive the main results. In addition, the use of regularization through a logarithmic barrier function, combined with an adaptive learning rate schedule, can also be considered a technically innovative aspect.

The obtained results are appropriately compared with those from previous studies, as cited, and appear to indicate solid improvements. Furthermore, some techniques provided in this paper have the potential to be useful in future research, and the newly constructed framework is designed to be accessible and applicable for subsequent studies (e.g., as demonstrated in Appendix A). I believe these contributions will be of interest to the online learning community. Based on these considerations, I support the acceptance of this paper.

**Paper Award:**

No

---

> ### Author Response · Authors · 2024-11-24
>
> Thank you for your review of our paper.

---

### Official Review · Reviewer_Q336 · 2024-11-10
**Comprehensive work with significant improvement for a very specific problem. Therefore, borderline for acceptance.**

**Rating:** 6
**Confidence:** 3

**Review:**

The paper addresses data-dependent regret bounds for Online Portfolio Selection, moving beyond standard worst-case regret. For the standard online scenario, the authors propose the LB-AdaCurv ONS algorithm, which achieves O(min(nR log(T),\sqrt{nT log(T)})) static regret (with \epsilon set to 0 or 1), where R is an unknown data-dependent parameter. The additional computational complexity per round is O(n^3).
In the second scenario, where the algorithm has a prior belief about the loss function (or future market trend) before each round, the authors propose the OUE-LB-FTRL algorithm. This algorithm achieves O(n log(T)) static regret when priors are accurate and O(\sqrt{nT log(T)}) static regret if the priors are arbitrary (worst case). Additionally, they introduce a meta-algorithm, Bob-OPS, which achieves O(log(T)) regret against an expected utility investor and O(n log(T)) static regret, both when the prior belief is accurate. Furthermore, they demonstrate that their algorithms can achieve dynamic regret for various dynamic scenarios.

Weaknesses:
Since it is a very specific problem (OPS) I think experiments are important here.
You write that the additional computational complexity of  OUE-LB-FTRL is O(n^3), however, it finds the minimization over the expectation of some distribution. How it can do it? Is it for specific distribution?

minor:
What is the \Psi presenting in the log-barrier regularizer?
I am unfamiliar with your names for the First-order/Second-order/Gradual Variation regret bounds. I know they are types of dynamic regret (or part of it). So, if those names/definitions are known it’s fine, otherwise it’s better to use known names/definitions.
Typos (not all):
In Alg.1 when you calculate M you write q\in\Delta. There is no q. You probably mean w. Also, the definition of the log-barrier regulaizer has typos.
I mentioned a few.

My wonders:
In LB-AdaCurv ONS there are two regularizers, both of which push to the equal element vector (all 1/n). what is the intuition for this? As we can see LB-AdaCurv FTAL has only one regularizer (\epsilon = 0).
As I understand, there is a connection between LT^* and VT. Thus, if it's true, you are supposed to be able to achieve the same Gradual Variation regret of the OUE-LB-FTRL algorithm by using the LB-AdaCurv ONS algorithm, but you don't do it. What is the reason is it not possible?

In conclusion, the paper presents new results for the OPS problem, which is a very specific problem. In my opinion, the proposed problem, results, and algorithms are of interest to the community, but I doubt that the work is significant enough to be published at the conference. Additionally, I mentioned a number of misunderstandings I had in the paper. I would appreciate the response of the authors.

#####
After reviewing the appendix, I concur that the work is broader than my initial claim. Thus, I revised my score to 6.

**Paper Award:**

No

---

> ### Author Response · Authors · 2024-11-24
>
> Thank you for your review of our paper. We address your questions below:
>
> Weaknesses: The computational complexity of OUE-LB-FTRL in general depends on the complexity of solving the stochastic optimization problem as it involves the minimization over the expectation of some distribution. The only place we claimed an $O(n^3)$ computational complexity for OUE-LB-FTRL (Theorem 9) was in Table 4 (page 6) regarding gradual variation regret bounds for online portfolio selection. In Theorem 9, we instantiate OUE-LB-FTRL with the logarithmic utility and delta distribution on $r_{t-1}$. Thus we only claim a runtime of $O(n^3)$ for this specific instantiation of OUE-LB-FTRL and not the general version. We will make this distinction explicit.
>
> Minor: We recognize that the the $\psi$ in the log-barrier regularizer $F_\psi$ is not necessary and can be safely removed, leading to better readability and notation.
>
> The names First-order/Second-order/Gradual Variation regret bounds are known in the literature. There are other names for these bounds, like Small-loss or $L^\star$ bound for First order bounds, Quadratic variation bound for Second order bounds and Path-Length bounds for gradual variation bounds. We will mention the various aliases for these bounds in the paper.
>
> Thank you for finding the typos that missed our scrutiny. We appreciate your effort in finding them and will correct them in the paper.
>
> There are two main difference between LB-AdaCurv FTAL and OUE-LB-FTRL. First is that OUE-LB-FTRL uses a hint function whereas LB-AdaCurv FTAL does not. If we set the hint function to 0, then OUE-LB-FTRL is just LB-FTRL, which only has linear terms plus LB regularizer. The LB-AdaCurv FTAL algorithm has both linear and quadratic terms plus LB regularizer. We found that the quadratic terms do not have any benefit  when adding hints. Thus we were not able to use LB-AdaCurv FTAL for obtaining gradual veriation regret bounds.
>
> In order to find new results for the OPS problem, we developed general OCO methods in our paper. These are mainly the results in Appendix A on OCO with Predicted Functions. The contents of this appendix are fully general and can be used by the broader online online convex optimization community. Thus we posit that our results have broader applicability than just OPS.

---

### Meta-Review · Area_Chair_ZTwT · 2024-12-06

**Recommendation:** Accept
**Confidence:** 5

**Metareview:**

This paper studies the fundamental online portfolio problem and develops new techniques and algorithms for getting various interesting data-dependent regret bounds. It should be a clear accept. Please do take the reviewers' minor comments into account when preparing the final version.

**Paper Award:**

No